# Connectomics of the *Octopus vulgaris* vertical lobe provides insight into conserved and novel principles of a memory acquisition network

**Flavie Bidel[1]\*[†], Yaron Meirovitch[2]\*[†], Richard Lee Schalek[2], Xiaotang Lu[2], Elisa Catherine Pavarino[2], Fuming Yang[2], Adi Peleg[2], Yuelong Wu[2], Tal Shomrat[3], Daniel Raimund Berger[2], Adi Shaked[1], Jeff William Lichtman[2], Binyamin Hochner[1]\***

[1]Department of Neurobiology, Silberman Institute of Life Sciences, The Hebrew University, Jerusalem, Israel; [2]Department of Molecular and Cellular Biology, Harvard University, Cambridge, United States; [3]Faculty of Marine Sciences, Ruppin Academic Center, Michmoret, Israel

**\*For correspondence:**
bidel.flavie@gmail.com (FB);
yaron.mr@gmail.com (YM);
benny.hochner@mail.huji.ac.il
(BH)

[†]These authors contributed
equally to this work

**Competing interest:** The authors
declare that no competing
interests exist.

**Reviewing Editor:** Albert
Cardona, University of
Cambridge, United Kingdom

**Abstract** Here, we present the first analysis of the connectome of a small volume of the *Octopus vulgaris* vertical lobe (VL), a brain structure mediating the acquisition of long-term memory in this behaviorally advanced mollusk. Serial section electron microscopy revealed new types of interneurons, cellular components of extensive modulatory systems, and multiple synaptic motifs. The sensory input to the VL is conveyed via~$1.8 \times 10^6$ axons that sparsely innervate two parallel and interconnected feedforward networks formed by the two types of amacrine interneurons (AM), simple AMs (SAMs) and complex AMs (CAMs). SAMs make up 89.3% of the~$25 \times 10^6$ VL cells, each receiving a synaptic input from only a single input neuron on its non-bifurcating primary neurite, suggesting that each input neuron is represented in only~$12 \pm 3.4$ SAMs. This synaptic site is likely a 'memory site' as it is endowed with LTP. The CAMs, a newly described AM type, comprise 1.6% of the VL cells. Their bifurcating neurites integrate multiple inputs from the input axons and SAMs. While the SAM network appears to feedforward sparse 'memorizable' sensory representations to the VL output layer, the CAMs appear to monitor global activity and feedforward a balancing inhibition for 'sharpening' the stimulus-specific VL output. While sharing morphological and wiring features with circuits supporting associative learning in other animals, the VL has evolved a unique circuit that enables associative learning based on feedforward information flow.

## Editor's evaluation

In this important study, neural circuit architecture within a brain module for learning and memory in the octopus was mapped from volume electron microscopy. The acquisition of this pioneering data set was followed by a compelling analysis of the circuits supporting learning and memory, and therefore behavioural plasticity, on the basis of the newly mapped vertical lobe microcircuit and prior studies on long-term potentiation (LTP) in this animal. The data and findings establish an important point of comparison with analogous brain structures and memory mechanisms in other organisms, such as the vertebrate cerebellum and the arthropod mushroom body, and expand the set of known and mapped neural circuit architectures for learning and memory. The results will serve as the basis for further studies on the fascinating cephalopods and inspire further exploration in the design of artificial neural networks.

## Introduction

The advanced learning capabilities of cephalopod mollusks like *Octopus vulgaris* are comparable to those of some vertebrates (*Sanders, 1975*; *Mather, 1995*; *Zarrella et al., 2015*; *Schnell et al., 2016*; *Hanlon and Messenger, 2018*). As the last common ancestor of cephalopods and vertebrates existed more than 500 million years ago (*Packard, 1972*; *Vitti, 2013*), the octopus provides a unique opportunity to study the independent evolution of nervous systems subserving sophisticated cognitive abilities (*Hochner et al., 2006*; *Shomrat et al., 2015*; *Hochner and Glanzman, 2016*; *Turchetti-Maia et al., 2017*). The VL, the highest structure in the hierarchy of the octopus' central nervous system, is regarded as a high-order integrative center involved in visually evoked associative learning and memory acquisition (*Boycott and Young, 1955*; *Sanders, 1975*; *Young, 1979*; *Fiorito and Chichery, 1995*; *Hochner, 2010*; *Hochner, 2012*; *Hochner, 2013*; *Shomrat et al., 2015*; *Shigeno et al., 2018*; *Turchetti-Maia et al., 2017*). It contains $25 \times 10^6$ neurons, about half the neurons in the central nervous system (*Young, 1963*). This number is not much smaller than the number of neurons in the human hippocampus ($\sim40 \times 10^6$, *West and Gundersen, 1990*) and about four orders of magnitude greater than the number of neurons in the mushroom body of *Drosophila* (e.g. *Aso et al., 2014*) and three orders of magnitude greater than that in the locust mushroom body (*Jortner et al., 2007*).

Extensive neuroanatomical data on the VL were provided by the work of J. Z. Young and E. G. Gray in the 1960–70's using light and electron microscopy (*Gray and Young, 1964*; *Gray, 1970*; *Young, 1971*). Based on estimated neuron numbers, they proposed that the VL network is extremely simple, composed of only two types of morphologically distinguishable neurons: $25 \times 10^6$ small AM and $65 \times 10^3$ large efferent neurons (LN) (*Young, 1963*). The main sensory input to the VL is conveyed via $1.8 \times 10^6$ axons projecting from the median superior frontal lobe (SFL). These axons compose the SFL-VL tract that runs in the outer neuropil of the five VL lobuli. Each of these axons makes *en passant* synapses with a then unknown number of the AMs. These $25 \times 10^6$ AMs converge sharply at a ratio of about 400:1 to innervate the LNs, whose axons were assumed to be the only output of the VL. Young pointed out that the pattern of SFL innervation of the AMs resembles the connectivity matrix in other learning and memory networks like the insect mushroom body and the mammalian hippocampus (*Young, 1995*).

This gross connectivity scheme provided insights that guided more recent physiological studies on the VL. Physiological experiments in the brain-slice preparation (*Hochner et al., 2003*; *Hochner et al., 2006*) revealed that the *en passant* SFL-to-AM innervation is glutamatergic. It shows a robust activity-dependent NMDA-independent long-term potentiation (LTP) of glutamate release from the SFL presynaptic terminals (*Hochner et al., 2003*). The AM-to-LN connections are excitatory cholinergic synapses (*Shomrat et al., 2011*). Recently it was discovered that LTP at the SFL-to-AM synapse is maintained by a novel nitric oxide (NO)-dependent 'molecular switch' mechanism in which NO itself reactivates the enzyme producing NO, nitric oxide synthase (NOS) (*Turchetti-Maia et al., 2017*; *Turchetti-Maia et al., 2018*).

Although Gray and Young provided a detailed description of the main neuronal elements constituting the VL network, the inability to reconstruct individual neurites limited the possibility of providing a comprehensive circuit connectivity. Indeed, their observations reported unclassified cell types, thus suggesting that the VL network is composed of more than the three neuronal elements (SFL, AM, and LN). Previous findings from our group also hinted that the VL is more complex. First, intracellular recording from LNs revealed that some receive inhibitory inputs (*Shomrat et al., 2011*). Second, a recent wide-ranging histolabeling study suggested that the AMs may be divided into at least two groups, one cholinergic and the other GABAergic (Stern-Mentch et al., 2022).

To understand the octopus VL circuitry at a higher resolution, our current study utilized recent advances in the ability to elucidate synaptic wiring diagrams at the nanoscale level using electron microscopy (EM). Widely applied to various species and tissues this methodology has contributed much to understanding the organization of neural networks (*Kasthuri et al., 2015*; *Lee et al., 2016*; *Morgan et al., 2016*; *Ryan et al., 2016*; *Motta et al., 2019*; *Li et al., 2020*; *Macrina et al., 2021*; *Meirovitch et al., 2021*; *Bae et al., 2021*; *Veraszto et al., 2020*; *Witvliet et al., 2021*; *Shapson-Coe et al., 2021*; *Winding et al., 2023*; *Loomba et al., 2022*; *Karlupia et al., 2023*). Here, we sought to produce a synapse-level wiring diagram of a small volume ($260 \times 390 \times 27$ μm) of one lateral lobule of the *Octopus vulgaris* VL from serial EM by tracing out all pre- and postsynaptic partners of dozens of main afferents of the VL (SFL projecting axons) and AMs. Indeed, although we found that the gross

anatomical organization of octopus VL fits the previous anatomical and physiological model described above, the nanoscale analysis revealed new types of neurons and connections that profoundly deepen our understanding of the functional organization of the octopus VL and how it implements associative learning and memory.

First, our data show that the AM interneurons are divided into two distinct groups, in contrast to previously thought. The majority (~89.3%) of the VL cell bodies are SAMs that send a single non-bifurcating neurite into the neuropil. A small fraction (~1.6%) forms a group of newly discovered CAM with a branching dendritic tree. Second, we found that each SAM receives only one synaptic input from a single SFL axon. We propose that this synaptic connection may serve as a 'memory site,' as it shows a robust activity-dependent LTP following tetanization of the SFL axonal tract (*Shomrat et al., 2011*). Third, the SFL-to-SAM expansion format suggests that each of the $1.8 \times 10^6$ SFL neurons is represented extremely sparsely in only around 12 of the $22.3 \times 10^6$ SAMs. Fourth, in sharp contrast to the SAMs, the CAMs integrate dozens to hundreds of inputs, suggesting that these interneurons integrate broader incoming SFL activity. As the SAMs and CAMs converge onto the same output layer of LN processes, our findings suggest the VL is comprised of two interconnected feedforward networks - one excitatory, originating from the SAMs (each representing one SFL) and a second, likely inhibitory parallel feedforward network, originating from the CAMs (each representing several SFL axons). We demonstrate that distinct groups of SAMs and CAMs neurites fasciculate together to form a columnar organization in which each column constitutes a canonical microcircuit that may function as an 'association module,' associating sensory features transmitted by the SFL. Lastly, we describe an additional afferent to the VL and two extrinsic putative neuromodulation systems. These interact with the feedforward networks and thus are likely involved in regulating the activity state of the network and in conveying reward and punishment signals.

Although our volume comprises only a small part of the lateral lobule, the literature (*Gray and Young, 1964*; *Gray, 1970*), and our exploration of a small volume of a VL median lobule suggest that the synaptic motifs involved in the fast transmission network are reiterated across the entire VL. This putative consistency is aligned with physiological studies that suggest the principal synaptic connectivity properties are independent of the location in the VL (*Hochner et al., 2003*; *Shomrat et al., 2011*). Therefore, we hypothesize that the general network organization is conserved across the entire VL, while neuromodulatory systems and anatomical compartmentalization may provide different functional roles for different VL areas or lobuli.

## Results
### The VL dataset and analysis

The region of interest (ROI) lay in the lateral lobule of the VL (*Figure 1A*). We cut, collected, and imaged 892 serial sections (30 nm section thickness) at 4 nm/pixel to generate a traceable 3D image stack covering a volume of 260 × 390 × 27 μm (*Figure 1B*; *Figure 1—video 1*; 'connectome dataset' available at https://lichtman.rc.fas.harvard.edu/octopus_connectomes).

To classify the cells, we used morphological features (e.g. main neurite orientation, presence of varicosities, branching pattern), ultrastructural features (e.g. vesicle size, mitochondria shape, cytoplasm staining, organelles) and connectivity assisted by the 'blueprint' established by *Gray and Young, 1964*; *Gray, 1970*; *Young, 1971*. Chemical synapses were identified based on features common to invertebrate synapses - the presence of a cluster of synaptic vesicles or 'cloud' associated with an active zone at the presynaptic membrane (*Gray, 1970*; *Ryan et al., 2016*; *White et al., 1986*; *Zheng et al., 2018*; *Meinertzhagen, 2019*; *Witvliet et al., 2021*). Synapses were annotated if this accumulation of synaptic vesicles persisted across at least three sections (>90 nm). We classified synapses as 'fast chemical' if the presynaptic side contained synaptic vesicles with 30–60 nm in diameter and as modulatory synapses if it contained larger vesicles (>60 nm) (*Messenger, 1996*; *Leng and Ludwig, 2008*; *van den Pol, 2012*; *Witvliet et al., 2021*). Consistent with previous studies (*Gray and Young, 1964*; *Gray, 1970*), we did not observe any obvious postsynaptic specialization (i.e. postsynaptic density) at any chemical synapse, and we found no evidence of gap junctions. We defined a postsynaptic site when a presynaptic specialization accompanied by a cluster of synaptic vesicles was juxtaposed with the postsynaptic element.

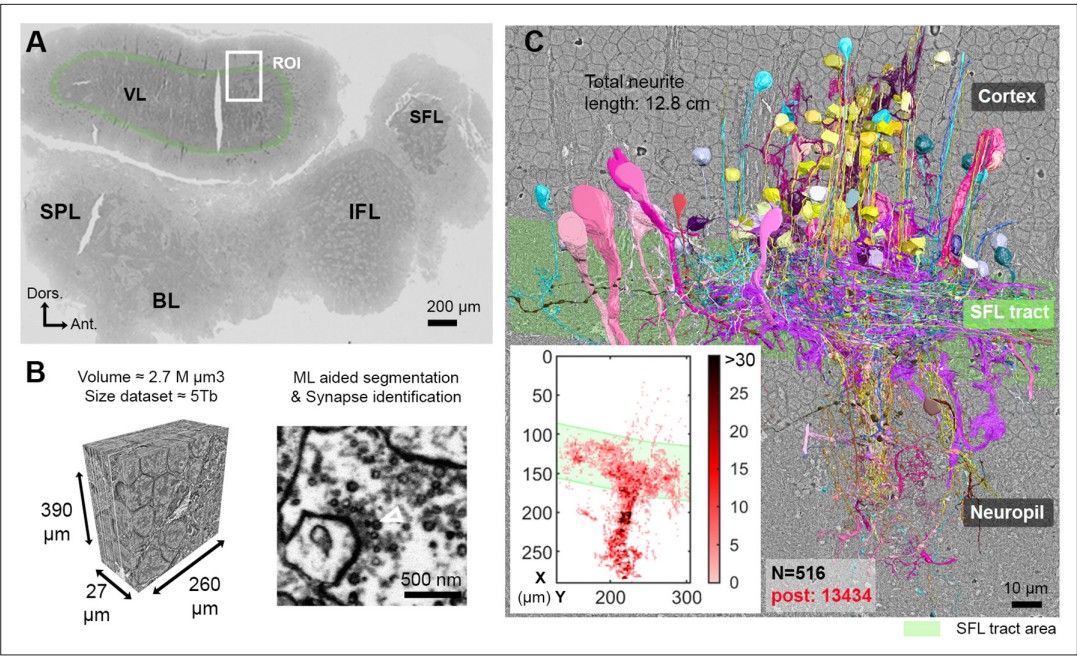

**Figure 1.** Reconstruction of octopus VL circuits using volume EM. (**A**) Low-resolution image of the octopus central nervous system. The VL lies dorsally in the supraesophageal part of the central brain. The region of interest marked within it (ROI, white rectangle) was scanned with a scanning electron microscope (SEM) at 4 nm/pixel. The approximate position of the SFL tract area (green) is shown for orientation and demarcating of the outer and the inner neuropils. (**B**) 30 nm sections were aligned in a traceable 3D stack for machine learning (ML) aided by manual segmentation and synapse annotation. (See also *Figure 1—video 1*). Chemical synapses were identified by the presence of a cluster or 'cloud' of synaptic vesicles (sv) associated with a pale active zone near the presynaptic membrane. The white empty triangle shows the cluster of sv within the presynaptic elements in the electron micrograph. Synapses were annotated if the accumulation of sv persisted through at least three sections in a row (90 nm). Consistent with previous studies (*Gray and Young, 1964*; *Gray, 1970*), none of the chemical synapses displayed a clear postsynaptic density (PSD). (**C**) The 516 reconstructed cell processes are colored according to cell type (See *Figure 1—animation 1* – for all cell types). Total neurite length of the reconstructed cells was 12.8 cm. Reconstructed cells are superimposed on a single EM image. Inset: spatial distribution of the 13,434 postsynaptic sites (post) annotated on the 516 cells presented as a heat-map. Abbreviations: BL, basal lobe; IFL, inferior frontal lobe; SFL, superior frontal lobe; SPL, subpedunculate lobe; VL, vertical lobe. See also *Figure 1—figure supplement 1* and *Figure 1—figure supplement 2*.

The online version of this article includes the following video and figure supplement(s) for figure 1:

**Figure supplement 1.** Three neural elements with yet unclear relationships with the vertical lobe (VL) connectome.

**Figure supplement 2.** Ground truth incorporated in the training dataset.

**Figure 1—video 1.** Vertical lobe electron microscopy (EM) volume (260 × 390 × 27 μm) imaged with 4 nm resolution in backscattered electrons mode.

https://elifesciences.org/articles/84257/figures#fig1video1

**Figure 1—animation 1.** Vertical lobe (VL) neural elements.

We reconstructed 516 cell processes with a total length of 12.8 cm. The reconstructed cell processes were classified into 10 cell types (*Figure 1—animation 1*) with 13,434 postsynaptic (*Figure 1C*, **heat map**) and 2472 annotated presynaptic sites. Among these 10 cell types, only seven were involved in the VL connectivity and, therefore, were analyzed in depth (see *Figure 1—figure supplement 1* for the three cell types not discussed here). For the characterization of the seven cell types, we reconstructed at least 30 processes per cell type throughout the volume and annotated all their pre- and postsynaptic sites. Additionally, a subset of cellular compartments and organelles (i.e. cell bodies, nuclei, and presynaptic varicosities) were separately segmented to allow morphological quantification. The core of our connectivity reconstruction included annotation and neurite reconstructions of all the synaptic partners of a subset of randomly selected 'seed' cells (similar to the approach taken

by *Morgan et al., 2016*): SFL axons (35 of the 84 reconstructed SFL axons) and SAM cells (28 of the 108 reconstructed SAMs).

We divided the analysis according to the three VL regions: (1) the outer cell body cortex; (2) the central neuropil, and (3) the SFL tract, or outer neuropil, where axons projecting from the SFL are arranged in a tract forming a ring around the outer zone of neuropil (*Gray and Young, 1964*; *Young, 1971*; Stern-Mentch et al., 2022). Note that because most of the cells are not fully contained in the reconstructed volume, it is impossible to assess the full collection of synapses between pairs of neurons. We, therefore, used 'synaptic site' to refer to a single anatomical release site or a single synapse.

## The VL connectome is characterized by two parallel and interconnected feedforward neural networks

*Figure 2* summarizes the seven identified neuronal cell types and their connectivity (see also *Figure 2—figure supplement 1*). None of the 84 reconstructed SFL axons bifurcated within the ROI (~260 µm long), suggesting a very low rate of bifurcation of the SFL axons, if at all, as suggested by *Young, 1971*. Each SFL axon innervated *en passant* two distinct sub-populations of AMs; a very large population of morphologically simple cells, which we termed SAM, and a small population of morphologically complex cells that we termed CAM (*Figure 2* and Figure 3).

The SAM and CAM cell bodies are intermingled in the cortex and their primary neurites run together in the same bundles, previously referred to as the *AM tract* (*Gray and Young, 1964*; *Gray, 1970*; *Young, 1979*). These bundles form a *quasi*-columnar organization toward the neuropil (Figure 6A and B and Figure 7A). The SAMs and CAMs differ in connectivity, morphology, and ultrastructure (Figure 6A and B). The SAMs are monopolar interneurons with a single non-bifurcating neurite. Their cell bodies are small (mean ± SD; 207.7 ± 26.1 µm$^3$; n=69) and contain almost no cytoplasm. In contrast, the CAMs have a complex bifurcating dendritic tree, a larger cell body (mean ± SD; 337.2 ± 41.6 µm$^3$; n=16), and cytoplasmic volume (Figure 4). Unexpectedly, we found that each SAM receives input from only one SFL axon (see below). In contrast, each CAM integrates synaptic inputs from a dozen to hundreds of SFL axons and SAMs (Figure 6B). Both the CAMs and SAMs consistently converged onto the LN processes (Figure 7). Overall, the transmission from the main afferents (SFL axons) is entirely mediated via the SAMs and CAMs to their shared target neurons - the dendritic processes of the LNs.

A second group of afferent axons innervated exclusively the CAMs. We termed these 'CAM-input-neurons' (CIN; Figure 8). Even though CIN function has yet to be established, their ultrastructure is reminiscent of the nerve endings which Gray and Young speculated may carry 'pain' signals to the VL. They described them as having neurofilaments ending shortly before the synaptic boutons, which were filled with small vesicles - a structure we also identified here. We also identified that the putative fast transmission circuit (panels framed with a solid line in *Figure 2*) is embedded in an intricate mesh of processes consisting of two types of putative neuromodulatory elements (NM and AF, panels framed with a dashed line in *Figure 2*), each with distinct morphology, ultrastructure, and cell-specific connections.

First, the 'neuromodulatory fibers' (NMs) manifest a large variety of processes with large varicosities filled with large vesicles (60–200 nm) that often do not show the typical structure of a synaptic connection (detailed below). Second, the 'ascending fibers' (AFs) climb from the neuropil up to the cortex through the AM tracts joining the dense bundle of primarily SAM and CAM neurites (Figure 9). A stereotyped feature of all AFs was their 'menorah'-shaped bifurcating pattern and their intense innervation by the SAMs. The ultrastructure and connectivity of these neurons revealed two distinct subcategories, one likely neuromodulatory (AF-Nm, Figure 9C) and the other as yet undefined (AF-Mix, Figure 9D). Both the AF-Nm boutons and the NM boutons were filled with large vesicles and usually lacked synaptic outputs, suggesting that these are release sites of neuromodulators or neuropeptides involved in extrasynaptic volume transmission (*Bentley et al., 2016*). Since none of the AF and NM neurites were associated with a reconstructed cell body, we hypothesize that AFs and NMs are external to the VL and may belong to the widespread ascending fibers entering the VL from below that have been previously described (*Young, 1971*).

Overall, the seven cell types form a feedforward neural network in the VL with fast synaptic transmission from one cell type to another, based on synaptic direction. The lack of connectivity 'loops'

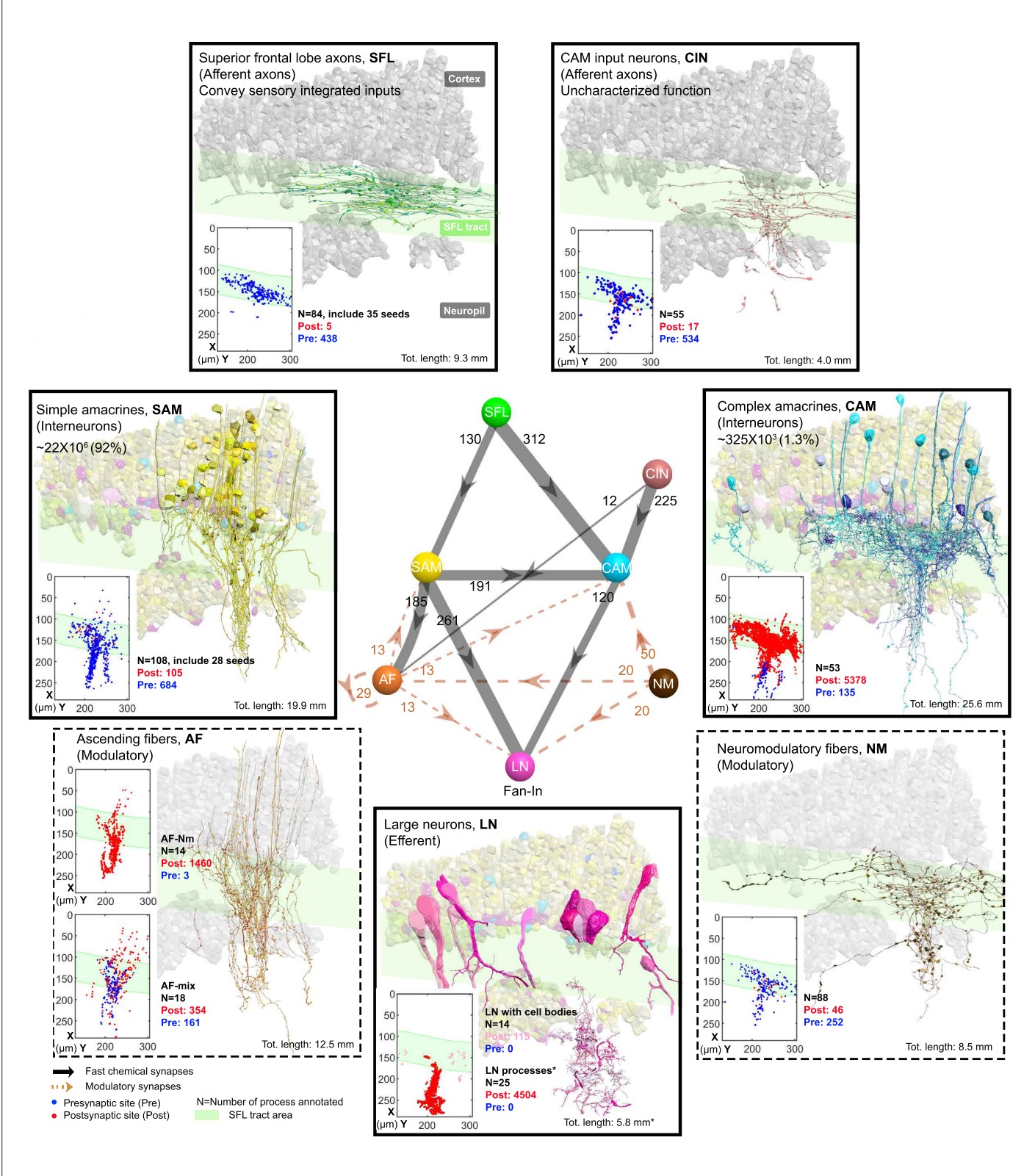

**Figure 2.** The seven neuronal cell types are classified in the region of interest (ROI) and their wiring diagram. The middle panel shows the wiring diagram, and the other panels depict the reconstruction of the seven neuronal cell types (superimposed on the vertical lobe (VL) cortex) together with the spatial distribution of their postsynaptic sites (red puncta) and presynaptic sites (blue puncta) within a portion of the ROI. Panels framed with a solid line represent neuronal elements involved in fast transmission (superior frontal lobe: SFL, CAM-input-neurons: CIN, simple AM: SAM, complex AM:

*Figure 2 continued on next page*

*Figure 2 continued*

CAM, large efferent neurons: LN). Panels framed with a dashed line represent neuromodulatory elements (ascending fibers: AF and neuromodulatory fibers: NM). The approximate position of the SFL tract area (green) is shown for orientation and demarcating the outer from the inner neuropil. In the central panel, each gray solid edge represents fast chemical synapses, and the arrowhead shows the direction of these synapses; line width indicates the relative number of synapses calculated from the number of connections annotated (noted next to each outgoing arrow). Orange dashed edges indicate chemical synapses observed with modulatory processes. Edges are given only if more than 10 synapses were annotated. The VL wiring diagram is characterized by two parallel and interconnected feedforward networks in which two afferents, the parallel SFL axons (green) that convey secondary visual features and a fiber that Gray speculated is associated with 'pain' (CIN-dark pink) innervate two distinct populations of amacrine interneurons, the simple amacrines (SAMs - yellow) and the newly discovered complex amacrines (CAMs - cyan). The large neuron processes (LN-magenta) provide the convergence element of the VL fan-in connectivity architecture and are the only VL output. Additionally, the circuit is supplied by a large network of likely modulatory processes: the 'ascending fibers' (AF-orange) and the neuromodulatory fibers (NM-brown). The connectivity among neurons in the VL is feedforward; neural activity is transmitted in one direction between populations of neurons without any feedback sub-circuitry among neurites (p<0.05 – see Methods, Analysis of feed-forwardness), except for a few bidirectional (reciprocal) synapses between AFs (modulatory). See *Figure 2—figure supplement 1* for neurite level analysis.

The online version of this article includes the following figure supplement(s) for figure 2:

**Figure supplement 1.** Wiring diagram at the neurite level (middle panel) surrounded by the panels of each of the seven cell types depicting their connectivity.

indicates forward transmission in the VL without feedback. The only possible exception was the AFs, which displayed several synaptic contacts (>10) with other AFs (AF-to-AF, *Figure 2*). Analysis at the single neurite level, including the entire wiring diagram, showed that the number of synapses involved in feedback circuitry in the ROI (i.e. cycles of information flow based on synaptic direction) was significantly smaller than chance (p<0.05). This was assessed from random wiring diagrams built from the same set of neurites and synapses as in our ROI but randomizing the directionality of the pre- and postsynaptic neurons for each pair of connected neurons (see Methods). The few feedback circuitry cycles found in the ROI were all associated with bidirectional (reciprocal) synapses between pairs of AFs, which themselves were rare (n=4). Note that extrasynaptic neuromodulatory transmission in the VL, which is not included in the connectivity analysis, could form feedback information flow orchestrated with the directional fast transmission network (see Discussion).

## Each SFL axon innervates the SAMs and CAMs via specialized bouton types

A set of 35 seed SFLs (afferents) was used for a detailed description of the connectivity pattern of the SFLs and their postsynaptic targets (*Figure 3A*). Each SFL axon (see one SFL axon segmented in *Figure 3—video 1*) was found to make two distinct types of synaptic contacts depending on the target interneuron. First, an average of 28 ± 12% (mean ± SD) of the synapses of each SFL axon (n=45/161, N=35 SFL) were made on SAMs (*Figure 3B* and *Figure 3—figure supplement 1*). This synaptic connection is unique as each SAM received only a single SFL input. The characteristic synaptic structure consisted of an especially large SFL presynaptic bouton (6.41 ± 2.1 µm³) innervating *en passant* a specialized vesicle-filled postsynaptic compartment protruding from the SAM primary neurite that partially wrapped the SFL LB (*Figure 3A*, *Figure 3—figure supplement 1* and Figure 4). We termed these postsynaptic structures 'palms' due to their somewhat flat and concave morphology. Second, an average of 66 ± 11% of the synapses (109/161 synapses; N=35 SFL) of each SFL axon were made on CAM dendritic endings. These CAM endings were a variety of twig-like structures formed by variably long (approximately 1–30 µm) thin protrusions emerging from thicker CAM dendritic branches (*Figure 3A*). These postsynaptic profiles were electron-lucent and filled with polyribosomes (*Figure 3B*).

Unlike the single SFL-to-SAM connection, one SFL axon could make several synapses on the same CAM. These SFL-to-CAM synapses showed two different arrangements. Either the synapses involved a large presynaptic bouton (LB) of the SFL, manifesting a multisynaptic glomerulus-like structure (*Figure 3A* and Figure 5), or the synapses involved only a single CAM output with a significantly smaller presynaptic SFL bouton (SB, 0.95 ± 0.88 µm³; *Figure 3A*). The volume of all boutons (N=295) of all reconstructed SFL axons (N=84) showed a bimodal distribution corresponding to these two different bouton types (*Figure 3C*).

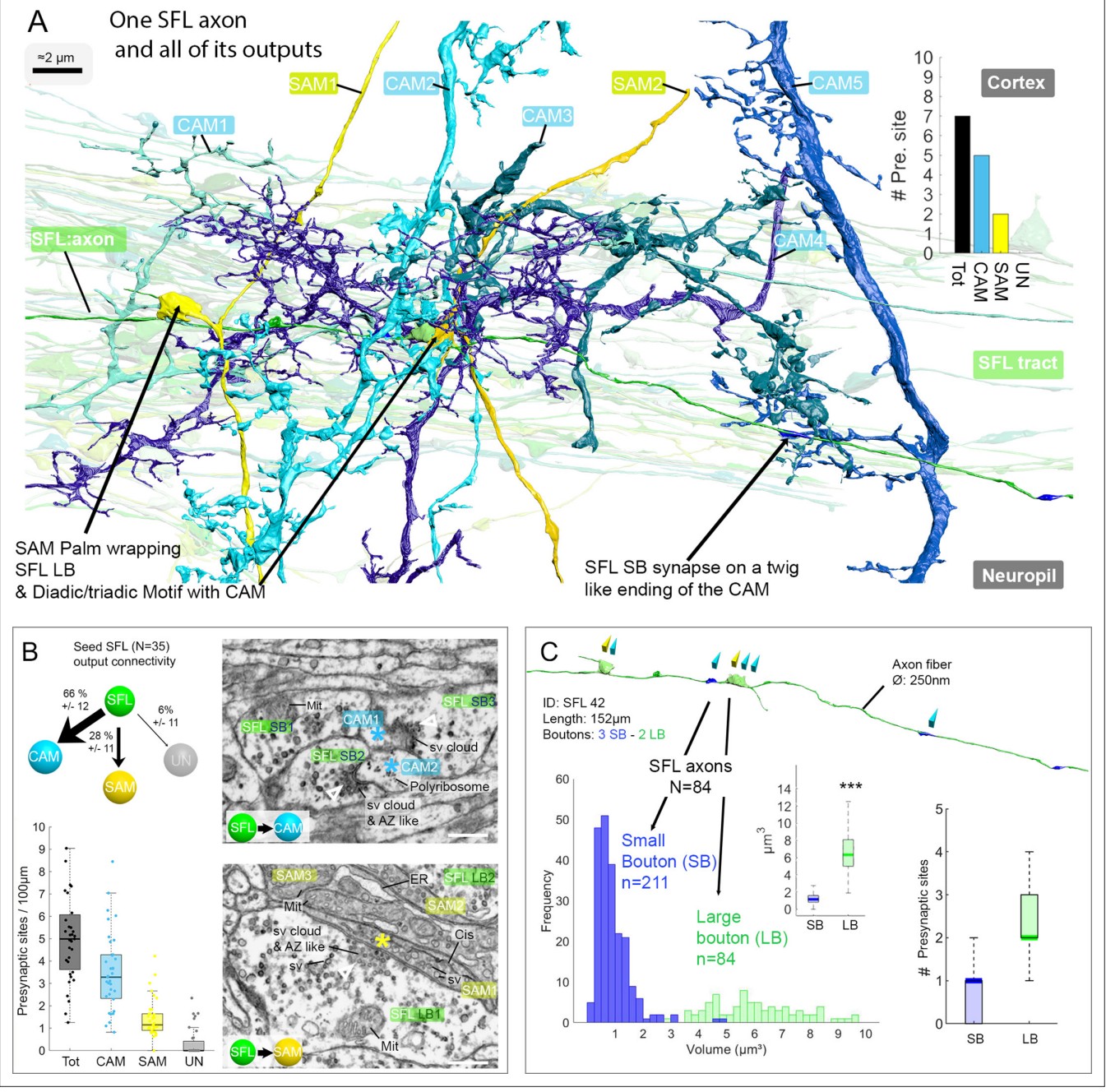

**Figure 3.** The SFL axons synapse onto the two populations of amacrine interneurons (simple AMs: SAMs and complex AMs: CAMs) via two distinct populations of presynaptic boutons. (**A**) Reconstruction of a single SFL axon (green) and its seven *en passant* outputs within the reconstructed volume: five CAMs (various blue hues) and two SAMs (yellow) superimposed on the 84 reconstructed SFL axons. (**B**) Quantification of the synaptic outputs of the seed SFL axons (n=35) revealed two main targets, the SAMs (via large boutons - LB) and the CAMs (via small boutons - SB), and a small portion of connections onto unidentified cell types (UN). Upper left shows the distribution of connections in percent. The box plot bottom left shows the number of different presynaptic sites along the 35 SFL axons (per100µm). On the right are high-resolution EM images of the two main SFL synaptic outputs onto SAM and CAM. 1 nm/pixel, dwell time: 200 µs. Scale bar, 500 nm. Empty triangle and asterisk are respectively pre- and postsynaptic profiles. The SFL axons synapse onto the CAMs via their small boutons and large boutons, and onto the SAM exclusively via their large boutons. (**C**) Example of the distribution of the presynaptic outputs of an SFL axon. This SFL axon synapses onto two SAMs (yellow triangles) and five CAMs (light blue triangles). The histogram displays the bimodal distribution of bouton volumes of 84 SFL axons into the two populations of SFL boutons: small (small boutons: SB-blue) and large (large boutons: LB-green). See the box plot showing the significant difference in volume between the SB (associated only with the CAMs) and the LB (associated with at least one SAMs palm and often involved in dyadic or triadic arrangement with a CAM). The box plot of presynaptic sites per SB and LB reveals their monosynaptic and multisynaptic nature, respectively. Asterisks indicate statistical significance ***p<0.001,

*Figure 3 continued on next page*

*Figure 3 continued*

\*\*<0.01. Abbreviations: sv, synaptic vesicles; AZ-like, active zone-like; Mit, mitochondria, Cis, cisternae; tER, transport endoplasmic reticulum. See also *Figure 3—figure supplement 1* and *Figure 3—video 1*.

The online version of this article includes the following video and figure supplement(s) for figure 3:

**Figure supplement 1.** The superior frontal lobe (SFL) axons form en passant connections at their large boutons with the SAMs' palms in a perpendicular arrangement.

**Figure 3—video 1.** One segmented superior frontal lobe (SFL) axon tracked within vertical lobe electron microscopy (EM) volume (260 × 390 × 27 µm) that was imaged with 4 nm resolution in backscattered electrons mode; a 3.2 nA beam current at 7 kV incident electron energy, and a dwell time of 200 ns/pixel were used.

https://elifesciences.org/articles/84257/figures#fig3video1

## The number of CAMs is extremely low relative to that of SAMs

We were next interested in the relative abundance of the SAMs and CAMs. Since we found significant differences in the volume and shape of the various cell types, we reconstructed 1482 cell bodies without their projections and used volume and shape descriptors to build a Support Vector Machine classifier to estimate the number of different types of neurons in the VL (*Figure 4*).

We estimated that 89.3% of the cell bodies in the sampled volume were SAM cell bodies and 1.6% were CAMs (*Figure 4A*) with an error rate on validation sets smaller than 4.01% (see Methods). Assuming that the VL is roughly composed of 25 × 10⁶ cell bodies (*Young, 1963*), the above

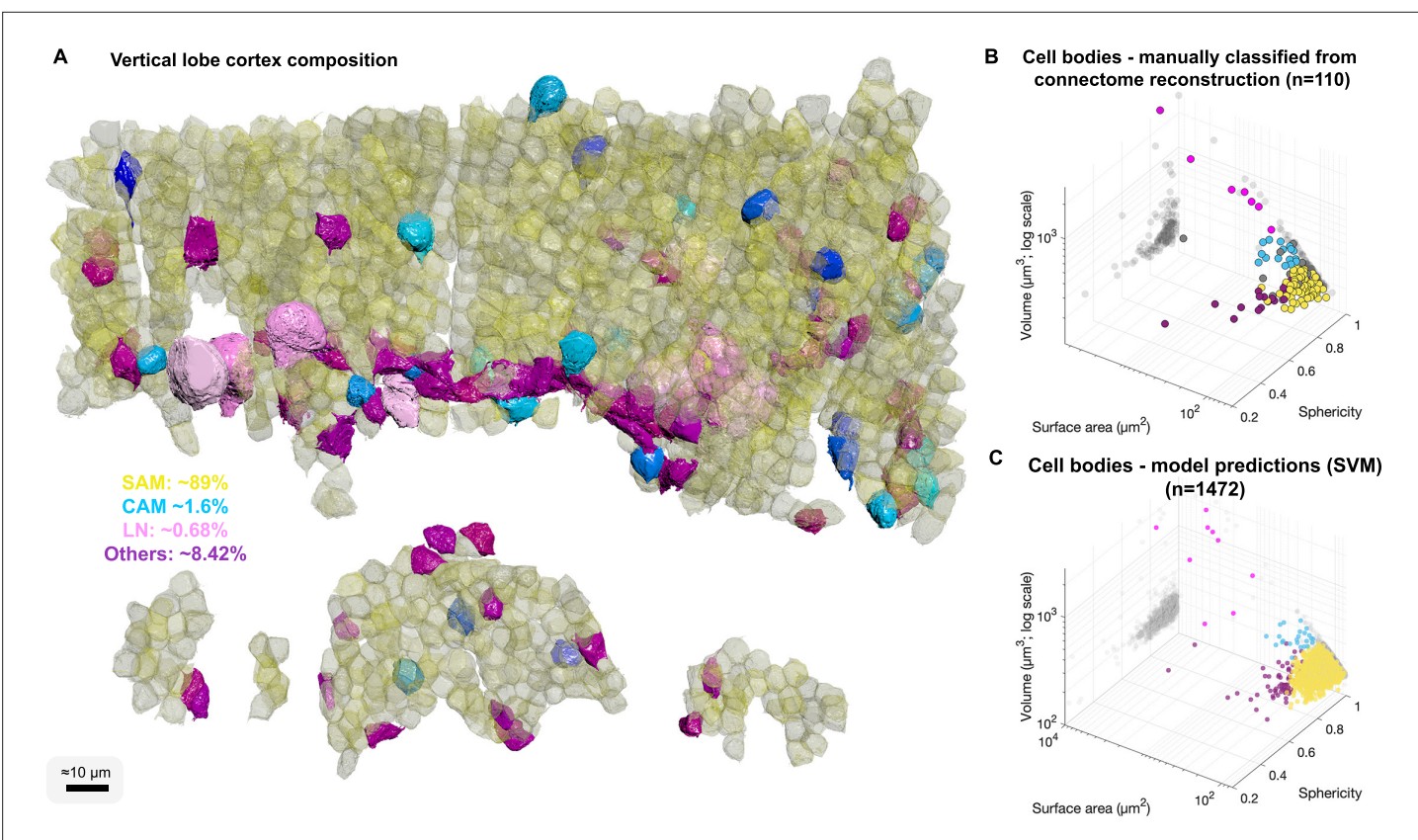

**Figure 4.** Cell body composition of vertical lobe (VL) cortex. (**A**) A subset of 1472 densely packed cell bodies were reconstructed and automatically classified based on their morphology; simple AM: SAM, yellow; complex AM: CAM, blue hues; large efferent neuron : LN, pink; Others, purple. (**B, C**) The classifier was designed to capture an observed correlation between cell type and cell body geometry. Roundish, relatively smaller cell bodies were associated with SAMs (vol 207.7 ± 26 µm³, mean ± SD). Larger and slightly more elongated cell bodies were associated with CAMs (vol 337.1 ± 41.6 µm³). Considerably larger cell bodies were associated with highly variable volumes with LNs (volume range 464–2850 µm³). Elongated cell bodies were associated with other cell types including glia and uncharacterized cells. A Support Vector Machine (SVM) was trained to separate four cell categories SAM, CAM, LN, and Others on a set of 110 manually classified cells based on three features: volume, sphericity, and surface area, and was validated on an independently prepared set with 47 cells (94.55% training accuracy; 4.01% error on the validation set, all associated with SAMs).

proportions extrapolated over the VL give an estimate of ~22.3 ± 1.2 × 10⁶ SAMs and an estimate that the cortex contains far fewer than $1.4 \times 10^6$ CAMs. As we did not observe any false negatives or false positives associated with CAMs, the total number is likely close to the classifier's best estimate of $400 \times 10^3$ CAMs, which can be further analyzed once this sparse set of cells is sampled in larger EM volumes. Assuming the SFL axons do not bifurcate in the SFL tract, then due to the one-to-one SFL-to-SAM innervation pattern, the $1.8 \times 10^6$ SFL axons fan-out onto the SAMs at an expansion ratio of about 1-to-12. Since each SAM is activated by a single SFL axon, the information from one SFL axon is represented at the SAM input layer in a non-overlapping manner in only about 12 ± 3.46 SAMs (assuming a Poisson distribution with an average of 12 LB).

To estimate the distribution of LBs along the SFL axons, we calculated the number of LBs per 100 µm in the 84 reconstructed SFL axonal segments. The results indicate 1.03 ± 0.73 LBs (mean ± SD) per 100 µm long segments suggesting highly sparse and variable distribution along the SFL axons. If this distribution is maintained along the axon, the 12 LBs would stretch over an average axonal length of 1262 µm, which grossly fits the length of the SFL tract in our preparation (*Figure 1A*), thus suggesting that the LB outputs into the 12 ± 3.46 SAMs are maintained at the same distribution sparsity along the entire length of the SFL axon, rather than being unevenly clustered at specific areas.

## The SAMs, CAMs, and SFL large boutons are interconnected within synaptic glomeruli

The SAM-to-CAMs connection forms the interconnection between the two parallel feedforward neural networks (*Figure 2*). These connections were found locally at a single large synaptic glomerulus encapsulating the SFL LB, the SAM palm, and the CAM twig endings (*Figure 5*). Within this glomerulus, these three neurons were repeatedly connected with multisynaptic motifs: monad, SFL-to-SAM (*Figure 5A*); dyad, SFL-to-CAM and SFL-to-SAM (*Figure 5B*), serial synapses, SFL-to-SAM-to-CAM, and triad where the same SFL presynaptic terminals innervated an adjacent SAM palm and a CAM twig ending (*Figure 5C*), as well as higher-order polyads.

This arrangement fits the organization of synaptic glomeruli in which a number of synapses are congregated in spatially compact structures, so that they may mutually affect each other's function (*Morgan and Lichtman, 2020*). Among these arrangements, the SAM palm, packed with vesicles, appears to be a unique bifunctional pre- and postsynaptic compartment, as most of the SAM palms formed serial synaptic connections (82%), receiving a synapse from a single SFL bouton and providing a synapse to 1–3 postsynaptic compartments, often twig endings of distinct CAMs. Analyzing a large population of 'long' SAM neurites (SAMs whose primary neurite was fully contained in the SFL tract; N=56) revealed that each SAM exhibited only a single palm along its inwardly directed primary neurite

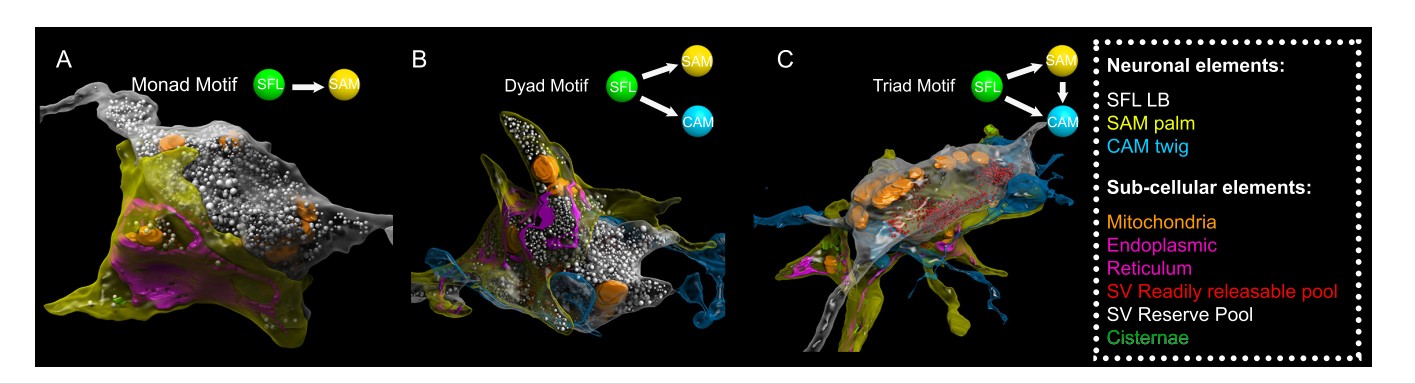

**Figure 5.** Superior frontal lobe (SFL) large boutons (LBs) are found in multisynaptic motifs forming a synaptic glomerulus involving serial connectivity with both simple AMs (SAMs) and complex AMs (CAMs). Subcellular 3D reconstruction illustrating the main motifs involving the large varicosity of the SFL axon, which contains glutamate and makes *en passant* synapses. (**A**) monad (**B**) dyad, and (**C**) triad. Note that in (**C**) only the presumably 'readily releasable pool' of synaptic vesicles is shown for clarity as otherwise the large 'reserve' pool in the SFL LB (>5000 synaptic vesicles) would obscure other components of the image. The SFL LB was always partly wrapped by at least one SAM palm (yellow), and they often additionally synapsed onto a twig-like postsynaptic CAM ending (blue).

(N=46/56), rarely two palms (N=2/56) (Figure 6C) and each palm received input from only one SFL large presynaptic bouton (50/50 palms received a single SFL input).

To better understand the structural properties of this unique palm structure, we prepared a smaller dataset of 16 × 16 × 3.5 µm from a median lobule and imaged it at 1 nm/pixel (high magnification dataset; see Methods). This higher resolution stack revealed that the complex presynaptic structure of the SAM palm could contain up to 1500 synaptic vesicles (36.8 ± 7.7 nm), often tightly packed and resembling the 'dense-walled' vesicles described by *Gray and Young, 1964*. This compartment included numerous elongated mitochondria and a large network of electron-lucent tubular organelles (*Figure 5*). These organelles were at times small, disconnected units (cisternae, Cis) or large and continuous, likely part of the endoplasmic reticulum (ER). The main body of the ER usually faced the pre- and postsynaptic sites, possibly playing a role in vesicle recycling (*Ramírez and Couve, 2011*; *Weigel et al., 2021*). We also occasionally observed what appeared to be exocytosis at the ER membrane, suggesting the possibility of a Golgi-like extended role in the synaptic vesicle machinery. It is still not clear whether the Cis (up to 15 observed in one palm) were part of the larger ER network or were isolated entities that could, for instance, play a role in keeping vesicles close to the active zone, like synaptic ribbons (*Dowling et al., 1966*; *Parsons and Sterling, 2003*) or the T-bar in *Drosophila* neurons (*Scheffer et al., 2020*).

## The SAMs distribute their single SFL input onto the LNs, CAMs, and AF

We next focused on a detailed reconstruction and proofreading of 28 SAM seed cells (see one SAM segmented in *Figure 6—video 1*). As mentioned above, each seed SAM neuritic trunk received, *en passant,* only one SFL LB input (*Figure 6A*). The LB input was always located at the SAM palm (*Figure 5* and *Figure 6A1 and C*). Analyzing the neuritic trunks of SAM seed cells showed that 89.3% of the seed SAMs appeared to have no other inputs than the SFL LB.

The inputs from SAMs to other processes were spread along the length of the neuritic trunk. These synaptic outputs targeted three cell types: CAM, AF, and LN (see below and *Figure 6F*). As described by *Young, 1971* and confirmed here, many SAMs converged onto the dendrite-like processes of the LN, mostly within the central neuropil but occasionally at the level of the SFL tract (*Figure 6A4* and *Figure 6E*).

However, the SAM inputs to the LN processes represented only 35% of the SAMs' presynaptic sites (N=109/271). The remaining connections are described here for the first time. 25% of the SAM outputs targeted the CAMs, while 26% innervated the processes of the climbing AFs, including neuromodulatory processes. The SAM-to-CAM synapses were formed by a variety of swellings of the SAM neuritic trunk, containing clear, small vesicle-filled profiles synapsing onto clear postsynaptic profiles of the CAM twigs (*Figure 6A2*). This was mainly observed within the SFL tract and in the upper neuropil. In contrast, the SAM-to-AF synapses (*Figure 6A3*) were spread throughout the dataset, including the cortex. The postsynaptic element (AF) was a bouton often filled with dense-core vesicles (Figure 9). In the neuropil and the SFL tract, these synapses were arranged in a rosette-like structure in which 2–5 adjacent SAMs synapsed locally onto the same AF bouton (Figure 9). Strikingly, within the cortex, the SAM soma occasionally synapsed onto AF varicosities, forming a heterodox somato-neuromodulatory synapse (Figure 9A). We also found rare cases of other synaptic outputs to NM (N=2/271), unidentified processes (N=19/271), and only one case of recurrent connection of a SAM onto itself (N=1/271) (*Figure 6F*). Although not identified in the seed cell analysis, we occasionally observed reciprocal synaptic contacts between SFL and SAM, as well as AF and SAM within the dataset.

## Morphology and connectivity of the CAMs

The CAMs were not identified in earlier studies likely due to their sparseness in the VL relative to the huge number of SAMs. *Gray, 1970* and *Young, 1971* may have possibly confused their dendritic arborizations and electron-lucent ultrastructure within the SFL tract (*Figure 6A2, B1 and B2 and B3*) with those of the LN due to their structural similarities. To understand the potential functionality of these newly discovered interneurons within the VL circuitry, we annotated all the pre- and postsynaptic sites of 53 CAMs. Because the SFL, CINs, SAMs AFs, and NMs all have distinctive ultrastructure 'fingerprints' (e.g. cytoplasm staining, vesicle size, vesicle type, mitochondria shape; see Methods), we were mostly able to unambiguously predict the presynaptic partners of the reconstructed CAMs from the ultrastructure without additional tracing.

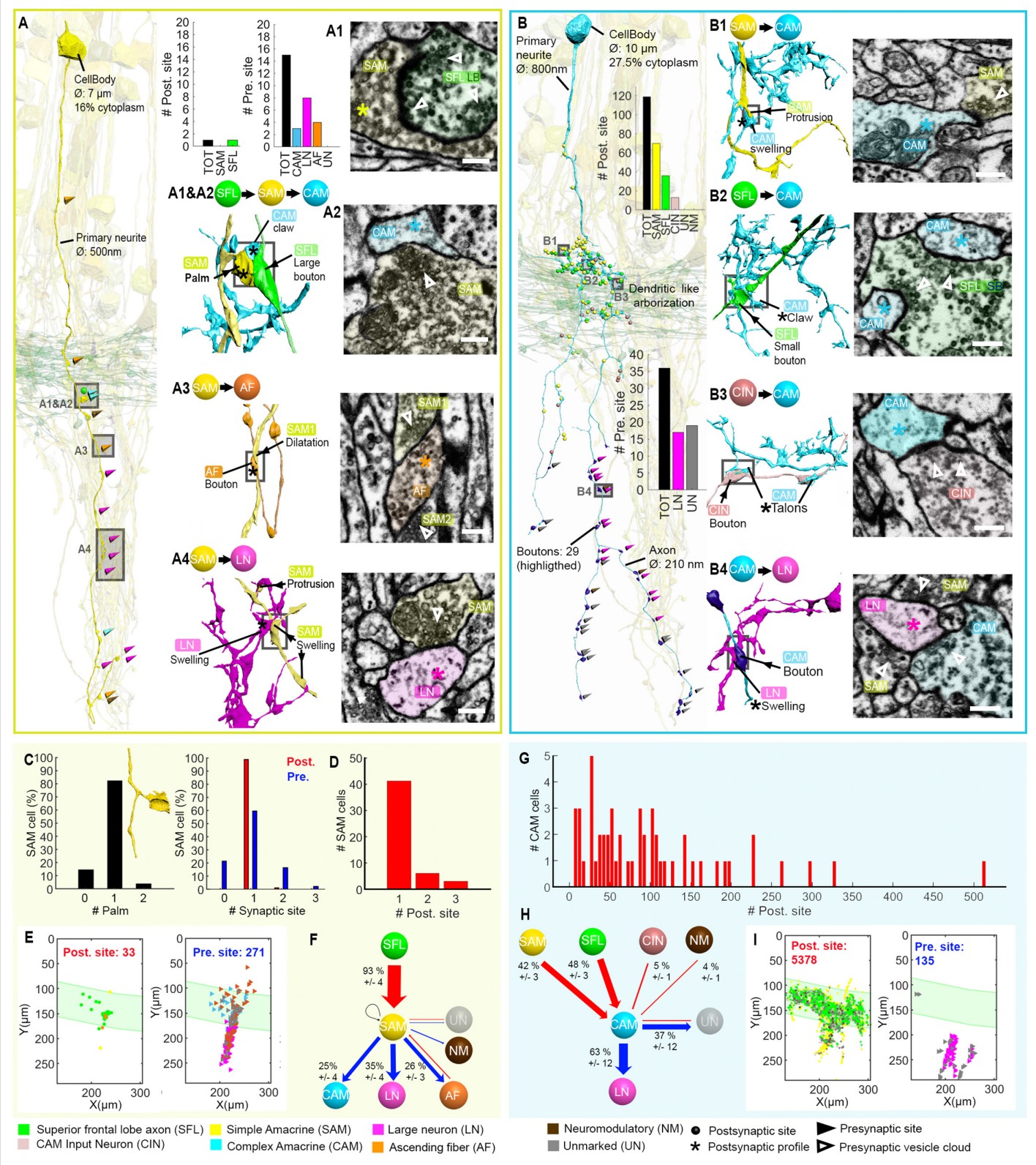

**Figure 6.** Stereotypic superior frontal lobe (SFL) ('mononeural')-input to the simple AMs (SAMs) and multiple ('polyneuronal') inputs to the complex AMs (CAMs). (**A,B**) 2D projection of a reconstructed SAM (A-yellow) and CAM (B-cyan), superimposed on a reconstruction of the SFL tract (green) and several SAMs (yellow). Triangles and puncta represent presynaptic output and postsynaptic input sites, respectively, color-coded according to pre- or postsynaptic partner. The insets show the corresponding electron microscopy (EM) cross-section of each synapse type. The morphological

*Figure 6 continued on next page*

*Figure 6 continued*

arrangement of the corresponding synaptic connection is depicted in the colored connectivity scheme next to each image and morphological reconstructions in the corresponding cells' color code (scale bar = 500 nm). (**C**) Analysis of the number of inputs onto the SAM 'palm' and its pre- and postsynaptic morphological specialization. 82% of SAMs had only one palm, each receiving just one input (red bar). Each SAM innervated up to three targets (blue bars). Only SAMs with their primary neurite fully contained in the SFL tract area were considered (N=56). (**D**) The majority of the SAMs had only one input located at the palm. (**E**) Spatial localization of the input/output of the 28 SAM seed cells within the ROI. Note the restriction of SFL-to-SAM inputs to the area of the SFL tract (green area). (**F**) The connectivity diagram of the 28 SAM seed cells shows the fraction of the inputs (red) and outputs (blue) by neuron type, revealing a single input type with the SFL (A1-green) and three main targets: the CAM (A2-cyan), ascending fiber (AF) (A3-orange), and large efferent neuron (LN) (A4-magenta). (**G**) Histogram showing the wide distribution of the number of synaptic inputs to the CAMs (dozens to hundreds), in contrast to the SAMs. (**H**) The connectivity diagram of 53 CAMs shows the percentage of total inputs to and outputs from the corresponding cell type. Most of the CAMs' presynaptic partners were not reconstructed but identified based on characteristic ultrastructure profiles. Profiles that could not be unambiguously identified were classified as unidentified (Un). Unlike the SAMs, the CAMs integrate dozen to hundreds of inputs from multiple SAMs (B1-yellow), SFL axons (B2-green), and CAM-input-neurons (CINs) (B3-dark pink). The CAMs appear to converge exclusively onto the LN (**B4**), the only postsynaptic partners identified. (**I**) Spatial distribution of the CAM inputs and outputs within the region of interest (ROI) showing the spatial segregation of the inputs (SFL tract mostly) and outputs of the CAMs (neuropil). See also *Figure 6—figure supplement 1*, *Figure 6—video 1*, *Figure 6—figure supplement 2*, and *Figure 6—figure supplement 3*.

The online version of this article includes the following video and figure supplement(s) for figure 6:

**Figure supplement 1.** Contrast between the single [mononeural] input of the simple AMs (SAMs) and the multiple [polyneural] inputs of the complex AMs (CAMs).

**Figure supplement 2.** Example of a vertically projecting complex AM (CAM) and its connectivity.

**Figure supplement 3.** A laterally projecting complex AM (CAM) and its connectivity.

**Figure 6—video 1.** One segmented simple AM (SAM) tracked within vertical lobe electron microscopy (EM) volume (260 × 390 × 27 µm) imaged with 4 nm resolution in backscattered electrons mode; a 3.2 nA beam current at 7 kV incident electron energy and a dwell time of 200 ns/pixel were used. https://elifesciences.org/articles/84257/figures#fig6video1

The CAM neurites have 3 distinct regions: the *primary neurite* ramifies into strictly postsynaptic *dendritic branches* in the SFL tract. The dendritic branches turn into neuritic *axons* in the deeper neuropil (*Figure 6B* and *Figure 6—figure supplement 2*). Synapses were located preferentially on numerous small twigs branching off the thicker dendritic branches. 'Twig' refers to the second-order dendritic branches of CAMs whose bases lie on a thick main branch containing microtubules (*Schneider-Mizell et al., 2016*), but, unlike spines whose maximal length is <3 µm (*Leiss et al., 2009*), the twigs can be much longer and show a complicated branching pattern. The CAMs showed variable dendritic branch patterns and orientation, possibly suggesting the existence of CAM subtypes (*Figure 6—figure supplements 2 and 3*). Additionally, the morphology of the CAM twig endings was highly variable with at least five categories, all somewhat resembling dendritic spine enlargements (*Figure 6B and C*): *flatfoot, long neck mushroom head, talon, palm-like*, and a *claw-like* ending that strikingly resembled the claw-like structure wrapping the projection neurons in the insect mushroom body (*Yasuyama et al., 2002*; *Leiss et al., 2009*; *Schürmann, 2016*).

In contrast to the SAMs (*Figure 6—figure supplement 1*), each CAM interneuron received dozens to hundreds of synaptic inputs (*Figure 6G* and *Figure 6—figure supplements 2 and 3*); the most extreme case displayed 514 postsynaptic sites over a neurite length of 1.4 mm. The CAMs integrate a combination of inputs (*Figure 6H*), mostly from the SFL (48%, *Figure 6B2* and *Figure 6—figure supplements 2G; 3E*) and the SAMs (42%, *Figure 6B1* and *Figure 6—figure supplements 2F; 3F*). 5% of the inputs to the CAMs originated from CINs (*Figure 6B3* and *Figure 6H*). The CINs appeared to exclusively innervate the CAMs with their inputs sparsely distributed over the CAM twig endings (Figure 8). Despite the CAMs showing high diversity in primary neurite projection and dendritic arborizations, all received inputs and innervated the same cell types and at similar locations within the VL (*Figure 6B* and *Figure 6—figure supplements 2 and 3*). The CAMs' distal axonal processes arborized and via, many boutons innervated, the LN processes *en passant* (*Figure 6B4*).

## Different convergence patterns of SAMs and CAMs onto the LNs

Both the SAMs and CAMs converged onto the LNs, the output cells of the VL (*Figure 2*, *Figure 6*, and *Figure 7*). To investigate the pattern of LN innervation, we first annotated all the synapses of 14 LNs with reconstructed cell bodies and those on putative LN processes (see below, N=24). The LN cell bodies (diameter >15 µm) and their radially directed large primary neurite were poorly contained

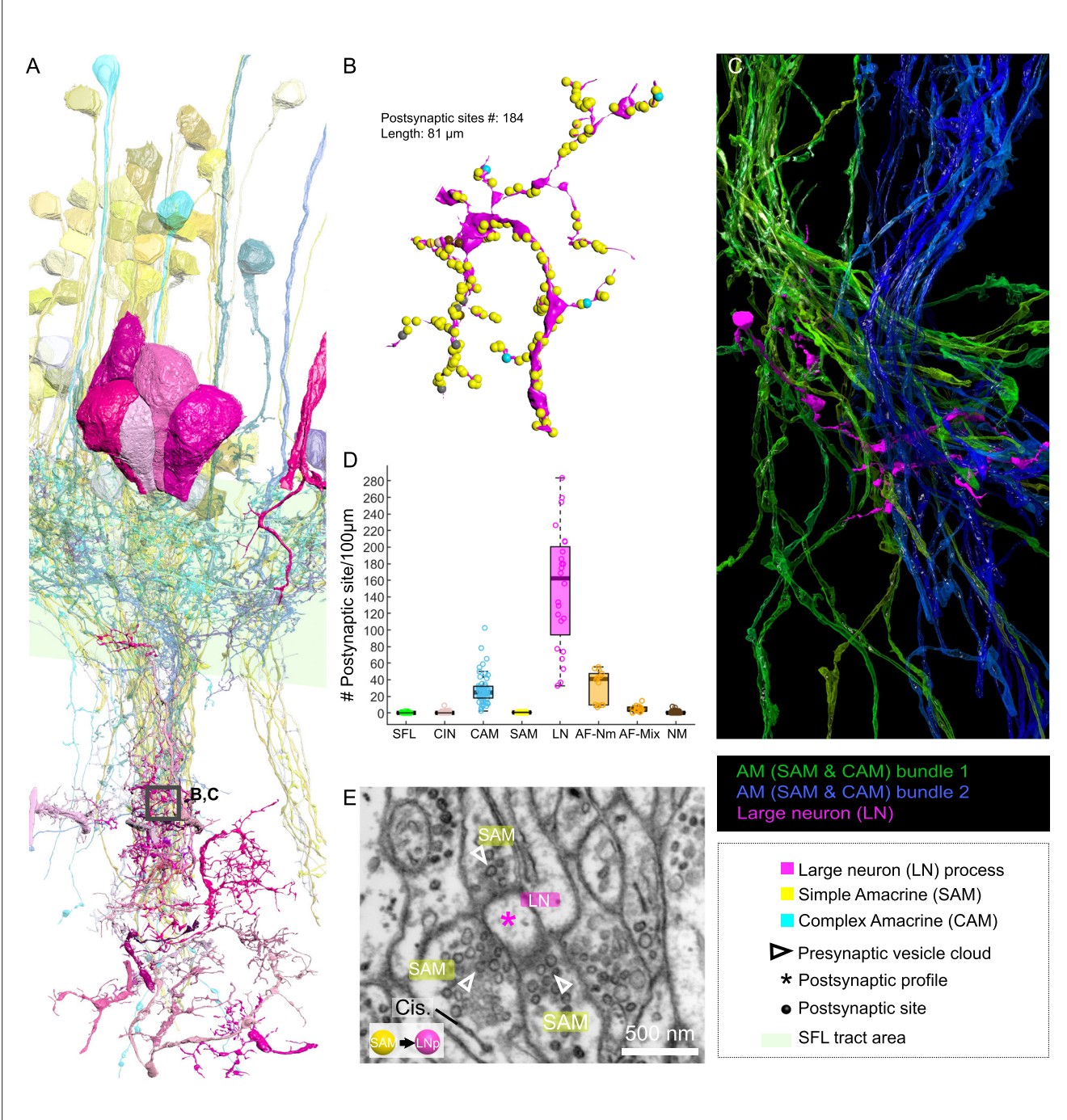

**Figure 7.** Patterns of synaptic convergence of simple AMs (SAMs) and complex AMs (CAMs) onto large efferent neurons (LNs). (**A**) 3D reconstruction of the LN cell bodies and processes (magenta hues) superimposed on SAMs (yellow) and CAMs (cyan). (**B**) Distributions of the 184 postsynaptic sites on one LN dendrite colored according to the key. Note the very few CAM inputs relative to those of the SAMs. (**C**) All partners of one LN neurite were reconstructed using a node-based agglomeration of 3D compartments (see Methods) to illustrate the convergence of two bundles of intermingled SAM and CAM neurites onto an LN process. (**D**) Box plot showing the density of postsynaptic sites for the seven neuronal elements of the VL. The LN processes displayed by far the greatest postsynaptic density among all neuronal elements. (**E**) High-resolution electron microscopy (EM) illustrating the typical synaptic motif (rosette-like structure) formed by the convergence of three SAMs into one LN process. Empty triangle and asterisk mark pre- and postsynaptic profiles, respectively. Resolution: 1 nm/pixel. Dwell time: 200 µs.

The online version of this article includes the following figure supplement(s) for figure 7:

**Figure supplement 1.** Three previous examples of whole cell LN recording that support the interpretations of the three types of connectivity pathways revealed by the current study (shown at the bottom of the respective **A, B and C** panels).

within our analyzed volume. However, some of them (N=3/14) gave off dendritic collaterals within the SFL tract area where they were solely postsynaptic to SAMs (*Figure 2*) and to other unidentified processes with a total of 115 individual synaptic inputs on the LN processes in the SFL tract.

Fitting the physiological results (*Shomrat et al., 2011*), we found no direct connections between SFL axons and LN dendritic collaterals. Among the seven identified cell types, we consistently observed that the processes of SFLs, SAMs, CAMs, CINs, AFs, and NMs all displayed compartments filled with vesicles within the neuropil. Therefore, by elimination, we considered all electron-lucent postsynaptic profiles within the neuropil as putative LN dendritic processes. The 24 examined LN processes branched profusely within the neuropil (*Figure 7A*) and were heavily innervated. The histogram in *Figure 7D* shows that the LN processes displayed the largest density of incoming synapses in the dataset with a median of 162 postsynaptic sites/100 µm, far larger than the densely innervated AF-Nms (41 postsynaptic sites/100 µm) and the CAMs (25 postsynaptic sites/100 µm that are restricted to the SFL tract; *Figure 7D*). The SAM and CAM innervation of the LN processes often formed rosette-like structures in which 2–5 or more AMs made adjacent synapses onto an LN dendritic process (*Figure 7E*).

We next estimated the degree of convergence of the SAMs and CAMs onto the LNs. For this, we reconstructed and annotated all the neurites synapsing onto an 81 µm length of an LN dendrite (*Figure 7B and C*). As shown in *Figure 7C*, all the CAM and SAM inputs to this LN originated from two distinct fasciculate neuritic bundles (blue and green) that included 160 SAM neurites and only six CAMs. Approximately 90% of the inputs on LN processes were from SAMs (N=160/184 postsynaptic sites), and only 3% were from CAMs (n=6/184 postsynaptic sites). *Figure 6B* shows that the CAM and SAM inputs were distributed in a 'salt and pepper' manner with the CAM inputs distributed very sparsely and evenly over the LN dendritic process. These data indicate that LNs (the VL output neurons) integrate massive information from the SAMs and only sparse input from the CAMs. The spatial distribution of the CAM inputs to the LN process suggests a role of the CAM output in adjusting the membrane potential of a large LN dendritic area. Such spatial distribution would be most effective if the CAMs are inhibitory interneurons as we suggest below (see Discussion).

## The CAM-input-neurons (CINs) exclusively innervate CAMs

CIN processes were spread throughout the neuropil, but many ran parallel to the SFL axons (*Figure 8A*). Their boutons were filled with many small, round mitochondria and the smallest vesicles observed in the VL (~30 nm). Their presynaptic boutons displayed the greatest variability in size, with volumes ranging from 0.09 to 29.50 µm$^3$ (median ± iqr; 1.65 ± 2.23).

Fifty-five of these fibers were reconstructed and their synaptic sites were annotated. The CINs were almost exclusively presynaptic to other processes (534 sites) with only very few postsynaptic sites (17 sites). All these presynaptic sites were located on large boutons (N=239) that usually made polyadic synapses with several CAMs' postsynaptic structures (e.g. three in *Figure 8B* and four in *Figure 8C*), with a median of 2.3 ± 1.5 (median ± iqr) synaptic sites per bouton (*Figure 8D*), the most extreme case being eight presynaptic sites for a single bouton. This synaptic structure may allow a single CIN terminal to influence several CAMs simultaneously.

The CINs innervated, *en passant* mainly the CAMs (225 synaptic connections) and less often the AFs (*Figure 2* and *Figure 2—figure supplement 1*). The CAMs occasionally projected long thin side branches with a morphological specialization at the tip (i.e. long neck with mushroom head, *Figure 8C*) which were innervated by the CIN bouton. The CINs displayed the highest presynaptic site density found within the VL, with a median of 11 presynaptic sites/100 µm, twice as high as presynaptic density on SFL axons (*Figure 8D*).

## The neuromodulatory AF receive massive synaptic inputs from the SAMs

The AF was highly recognizable in our dataset due to their 'menorah'-shape, climbing through the neuropil into the cortex while intermingling with SAM and CAM neurites (*Figure 9A*). Their processes displayed large boutons along their length (9.4 boutons/100 µm). The reconstruction of the synaptic partners of the SAM seed cells revealed that the AFs receive massive input from the SAMs (*Figure 6F*).

Mapping the AF synaptic sites revealed two clear subcategories of AFs based on their connectivity (*Figure 9B*) and EM profile (*Figure 9C and D*). We termed one class 'AF-neuromodulatory'

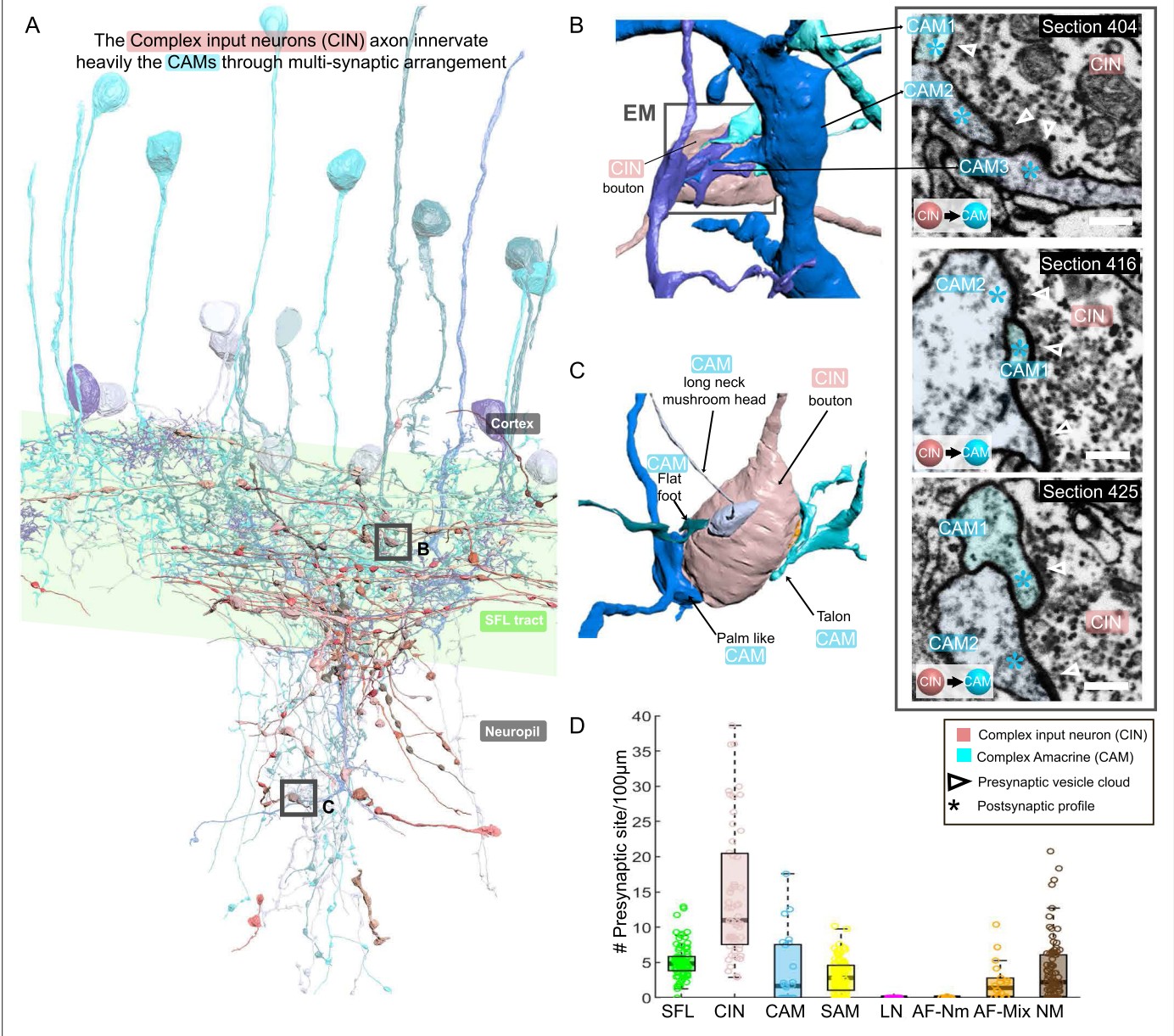

**Figure 8.** CAM-input-neurons (CINs) exclusively innervate complex AMs (CAMs) via large multisynaptic boutons. (**A**) Reconstruction of the CINs (pink) superimposed on the CAMs that are their main postsynaptic partners (blue hues). (**B, C**) 3D arrangement of a large CIN presynaptic bouton innervating multiple CAM dendrites (3 in **B**; 4 in **C**). CAM twig endings varied in shape resembling flattened dendritic spine enlargements. As indicated in (**C**), their structure can be divided into at least 5 classes. Right inset, three consecutives near cross-sections through a CIN bouton and its three postsynaptic targets. The CIN profile contained the smallest vesicles, about 30 nm in diameter, of the entire dataset. (**D**) Box plot showing the density of presynaptic sites for the seven neuronal elements of the ventral lobe (VL). CINs had the highest presynaptic site density of the dataset. Error bar = standard error of the mean.

(AF-Nm; n=14). In this class, on the 468 boutons reconstructed, we annotated 1460 postsynaptic sites and only three presynaptic sites, with a median of 3.2 postsynaptic sites per bouton. Each bouton formed a rosette structure of axo-axonal synapses where several SAMs made adjacent synapses onto one AF bouton (*Figure 9C*). The postsynaptic sites were filled mostly with large dense core vesicles (60–80 nm), suggesting a neuromodulatory nature. The synaptic inputs of the SAMs to the AF-Nm boutons suggest SAMs-induced local extrasynaptic release of neuromodulator (see Discussion). The AFs seemed morphologically equivalent to the serotonergic processes ascending into the cell cortex

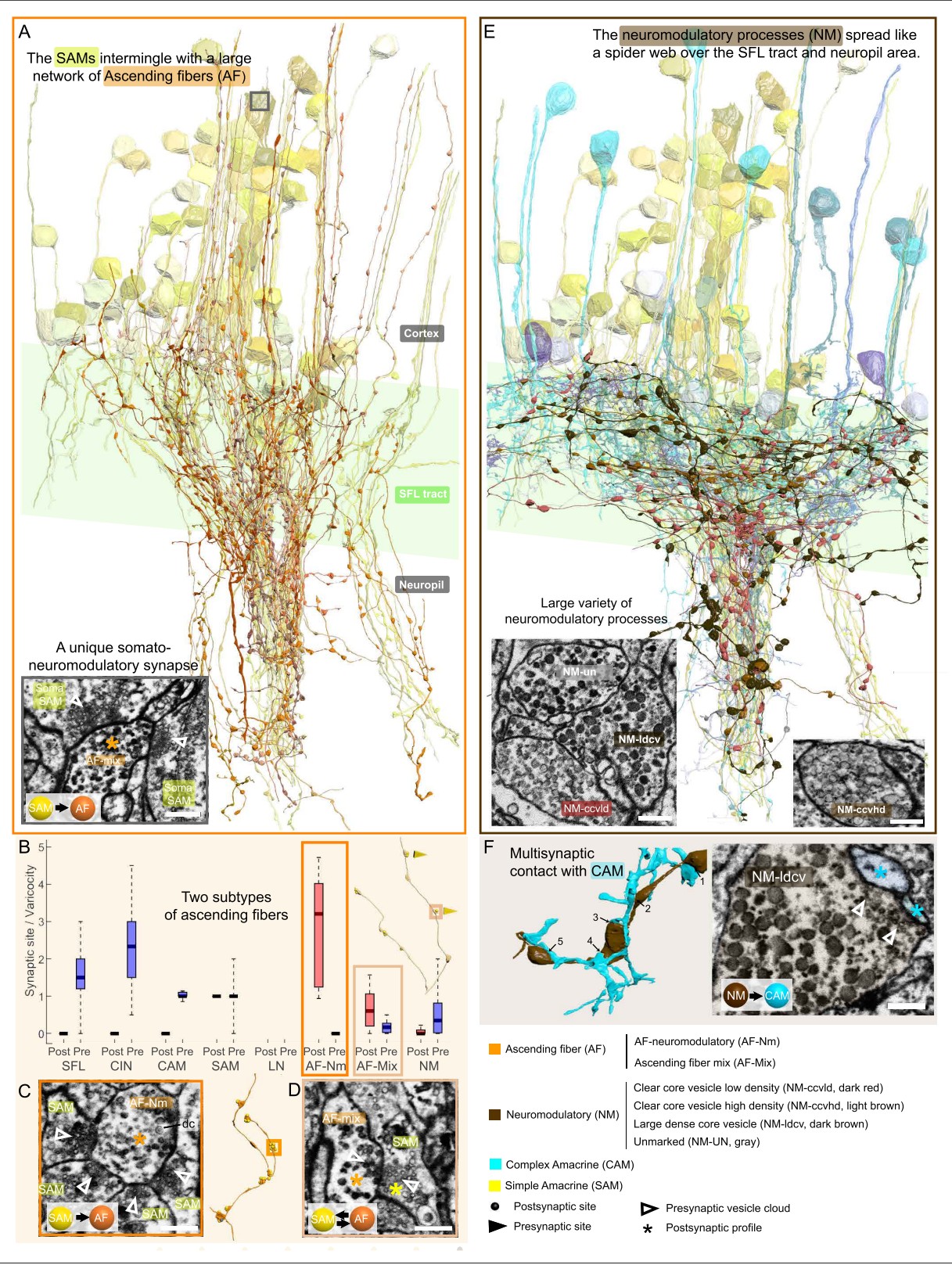

**Figure 9.** Overlay of two modulatory systems, the ascending fibers (AFs) and widespread neuromodulatory fibers (NMs). (**A**) 3D reconstruction of a large network of AFs (orange) intermingling with a SAM neuritic bundle (yellow). The AFs showed varicosities along their length filled with large vesicles and were mainly postsynaptic to the simple AMs (SAMs) (**C, D**) A somatic synaptic input from SAM to AF was occasionally observed (lower left inset). (**B**) The distribution of the number of pre- and postsynaptic sites per varicosity (blue and red, respectively) for the seven neuronal elements considered revealed

*Figure 9 continued*

two types of AF processes. (**C**) The AF-Nm type (orange) were exclusively unidirectional synapses forming a rosette-like structure where 2–5 SAMs made adjacent synapses onto one AF-Nm postsynaptic varicosity (asterisk). (**D**) The AF-mix (light orange) was more diverse. Their boutons were filled with vesicles of various shapes and sizes, ranging from small dark core vesicles to flat clear core vesicles. Unlike the AF-Nm, they could be bidirectional, both pre- and postsynaptic (triangles and puncta, respectively), and occasionally displayed reciprocal contact with a SAM as shown in the electron microscopy (EM). (**E**) 3D reconstruction of the widespread NM fibers superimposed on populations of SAMs (yellow) and CAMs (blue). At least three distinct subtypes were distinguishable based on vesicle sizes (EM insets: clear core low density, clear core high density, dense core). (**F**) 3D reconstruction of NM-to-CAM synapses. The NM occasionally made multisynaptic contacts. The right inset shows a representative EM of a neuromodulatory synapse. Scale bar EM sections, 500 nm. NM synapses were distinguished by the presence of much larger vesicles (>60 nm) in the presynaptic varicosities than those observed in traditional chemical synapses (fast transmission).

The online version of this article includes the following figure supplement(s) for figure 9:

**Figure supplement 1.** Reconstruction of one ascending fiber (AF)-mix (orange) and its inputs from the simple AMs (SAMs) (yellow).

found in immunohistological studies (*Shomrat et al., 2010*; *Shigeno and Ragsdale, 2015*) and also to tyrosine hydroxylase (catecholaminergic) labeled processes (Stern-Mentch et al., 2022).

In contrast, the second class of AF, the AF-mix (N=18), was more diverse with boutons filled with vesicles of varying shape and size (30–90 nm; N=519), including small dark core vesicles, flat clear core vesicles, and larger vesicles. Unlike the AF-Nm, the AF-mix was both presynaptic (median of 0.2 presynaptic sites/bouton) and postsynaptic (median of 0.6 postsynaptic sites/bouton) and occasionally displayed reciprocal contact with a SAM (*Figure 9D*). The diverse nature of the synaptic vesicles in the two subtypes suggests that they convey different neuromodulatory signals, and the reciprocal connection with the SAMs hints at their involvement in the SAM-induced local release of neuromodulators (see Discussion).

## The ubiquitous NM appear to be involved in global neuromodulatory signaling via extrasynaptic transmission

A large variety of processes displayed the largest dense vesicles (60–200 nm) in our dataset, similar to neuromodulatory vesicles in other invertebrates (*Messenger, 1996*; *Witvliet et al., 2021*). The processes were widespread within the neuropil and the SFL tract and their lack of specific orientation made it impossible to assume their origin (*Figure 9E*). At least three distinct profiles were clearly distinguishable by vesicle size and appearance, showing either a low density of clear core vesicles, a high density of clear core vesicles, or very large dense core vesicles. The boutons of each process contained mainly one type of vesicle, which were also present at a much lower density within the fiber. Reconstructions of NM processes (N=88) showed that most of the boutons had neither inputs nor outputs (464/632), suggesting that they were neuromodulatory afferents that exert their effects through extrasynaptic volume transmission. Yet, these cells occasionally synapsed with a variety of postsynaptic partners, including CAMs, SAMs AFs, NMs themselves, and LNs (*Figure 2* and *Figure 2—figure supplement 1*). Multisynaptic contacts were occasionally observed, for instance, with CAMs (*Figure 9F*), but the density of synaptic sites per bouton was rather low; 52 presynaptic sites and 250 postsynaptic sites on 632 boutons. Accordingly, we hypothesize that the NM processes are primarily involved in extrasynaptic transmission and thus may convey a global modulatory signal.

## Discussion

The novel connectivity and anatomical features discovered here profoundly advance our understanding of the functional organization of the learning and memory circuits in the VL of *Octopus vulgaris*. The EM volume reconstruction revealed several novel features that show the VL is composed of several more components than previously thought. These novel cell types and their connectivity help us understand the functional organization of the VL as a learning and memory acquisition network.

## The amacrine interneurons in the VL input layer are not a homogeneous group

The first striking finding was that about 89% of the ~25 × 10^6 VL cells (*Figure 4*) send a non-bifurcating neurite into the neuropil and were, therefore, termed simple AMs (SAMs; *Figure 6A*). The SAMs are highly input specific as each received *en passant* input from only a single SFL axon via large

presynaptic boutons (LB) (*Figure 3A*, *Figure 5*, and *Figure 6A1*). If the pattern of one-to-one innervation is maintained throughout the whole SFL tract and if, as generally believed and as found here, the SFL axons bifurcate very rarely, then every SFL neuron is represented by only around 12 ± 3.46 SAMs (22 × $10^6$ SAMs/1.8 × $10^6$ SFL axons). Moreover, the LBs seem to be sparsely distributed along the SFL axon (1.03 ± 0.73 LBs per 100 μm; see Results), suggesting that each SFL neuron is represented in the SAMs in a highly sparse and non-overlapping fashion, as suggested for the spacing of axonal varicosities in mammal's hippocampus and cerebellum (*Shepherd et al., 2002*; *Lanore et al., 2021*).

The finding that a single SFL input innervates each SAM is unprecedented for memory acquisition circuits like the cerebellum and the mushroom body (*Laurent, 2002*). These networks are organized in a fan-out (i.e. expansive) architecture in which low-dimensional sensory information is represented in a relatively small number of neurons in an input layer (e.g. different odors are represented in the ~50 glomeruli of the fly antennal lobe *Keene and Waddell, 2007*). To further process the sensory information, the neurons of the input layer project into a much larger number of interneurons in an intermediate or mixing layer (e.g. the 2000 Kenyon cells in the fly mushroom body), where the dense low-dimensional representation at the input layer is transformed into a high-dimensional and sparse representation (*Babadi and Sompolinsky, 2014*). The connectivity scheme that theoretically determines the optimal level of increase of dimension takes into account the number of neurons in the input layer relative to those in the intermediate layer (i.e. expansion ratio) and two quantitative interconnection parameters: (1) how many input neurons converge onto each neuron in the intermediate layer; and (2) how many neurons in the intermediate layer are innervated by a single neuron in the input layer (i.e. divergence) (see *Litwin-Kumar et al., 2017*). As each neuron in the intermediate layer responds to a (possibly random) combination of several inputs (e.g. Kenyon cells represent a combination of several (~6) odors *Li et al., 2020*), each neuron in the intermediate layer now represents a distinct multidimensional combination of several sensory features (e.g. a dimension that classifies a multi-odors object like a distinct food source). As the SAMs receive only one input, a priori, they cannot participate in such higher-order representation of the SFL input but merely expand it 12 times in the SAMs layer.

We, therefore, suggest that as each SAM represents just a single SFL axon, the SAMs may be viewed as relaying their SFL inputs directly into the LN and CAM dendritic processes. This suggests, a priori, that the SAMs, especially with their linear input/output relationship (*Shomrat et al., 2011*), are computationally redundant. However, the SFL LB-to-SAM synapse is also the site of short- and long-term synaptic plasticity. Thus, we propose that, while the SAMs appear architecturally redundant, they still may provide a dedicated synaptic site for scaling specific sensory input at the connection between the presynaptic large bouton of the SFL axon and the postsynaptic palm of the SAM neurite. Indeed, as we discuss below, our findings raise the possibility that ensembles of SAMs are organized in 'association modules' that enable them to participate both in processing sensory information and in forming combinatorial associations during learning.

Our second novel discovery is that the AMs included a small group of a new class of AM neurons, only 1.6% of the AMs in the VL cortex (*Figure 4*). These neurons showed a bifurcating dendritic structure and were, therefore, termed complex AMs (CAMs, *Figure 6B*). The primary neurite segments of the SAMs and CAMs travel together in the AM bundles within the VL cortex, but the non-bifurcating SAM neuritic trunks cross the SFL tract to pass into the inner neuropil, while the CAM neurites bifurcate substantially within the SFL tract (*Figure 2* and *Figure 6B*). The CAMs receive inputs *en passant* from multiple SFL axons either directly or indirectly via a serial connection from the SAMs (*Figure 5* and *Figure 6B*). Thus, while each SAM represents a single SFL neuron, the CAMs integrate information from dozens to hundreds of SFL neurons together with multiple inputs from nearby SAMs. In addition, as discussed below, the CAMs receive inputs from afferents from a yet unidentified origin, which exclusively innervate CAMs (CIN, *Figure 6B3* and *Figure 8*).

## The SFL axons innervate the SAMs and CAMs via different classes of presynaptic boutons

SAMs and CAMs are innervated by SFL axons via small (SFL SB) or extremely large boutons (SFL LB – *Figure 3*). The LBs are always associated with at least one SAM palm and more commonly innervate a CAM twig within a dyadic or triadic arrangement (*Figure 5*). By contrast, SBs are always associated with a single postsynaptic CAM compartment (*Figure 3*). This suggests that the two different types

of SFL axon terminals are functionally specialized. Such functional segregation is also seen in the hippocampal mossy fiber connections where small boutons target GABAergic interneurons, whereas large boutons contact CA3 pyramidal cells and hilar mossy cells (*Acsády et al., 1998*). The latter large synapse shares several synaptic features with the octopus VL synapses (*Hochner et al., 2003*) – NMDA-independent, presynaptically expressed LTP and an extremely large range of presynaptic short-and long-term synaptic gain modulations (*Nicoll and Schmitz, 2005*; *Henze et al., 2000*; *Hainmueller and Bartos, 2020*; *Restivo et al., 2015*). The large synaptic boutons here, like the mossy fiber terminals, may provide an increased dynamical range of synaptic gain control needed for certain forms of plasticity involving presynaptic modulation of transmitter release (*Orlando et al., 2021*; *Hainmueller and Bartos, 2020*; *Restivo et al., 2015*).

## The single SFL synaptic input into the SAM may serve as a gain control site of the stimulus-specific input to the VL

Each SAM receives synaptic input from only a single SFL neuron, apparently making them computationally redundant, as discussed above. This raises the question of the role of such apparently non-integrating interneurons. Physiological results indicate that the glutamatergic SFL-AM synapse is the site of activity-dependent LTP in the VL (*Hochner et al., 2003*; *Shomrat et al., 2011*; *Turchetti-Maia et al., 2018*). The large volume of the SFL-LB relative to that of the SFL-SB found here (on average sixfold larger) suggests that the recorded synaptic field potential (fPSP) mainly reflects the synaptic input from the SFL LB to the SAMs. The morphological dimensions of the SAMs revealed here (e.g. a 166 µm long 0.5 µm diameter neuritic trunk emerging from a 7 µm diameter cell body) together with previous electrophysiological measurements (>1 GΩ input resistance measured at the cell body), suggest that the inexcitable SAMs (*Hochner et al., 2003*; *Hochner et al., 2006*) are electrotonically compact (estimated length constant λ>147 µm *Hochner and Spira, 1987*), so that the glutaminergic excitatory postsynaptic potential from the SFL LB input site is likely sufficient to activate voltage-dependent transmitter release down the SAM neurite without needing a propagating action potential. These inexcitable properties, which differ drastically from the excitable properties of the insect Kenyon cells (e.g. *Demmer and Kloppenburg, 2009*), nicely explain the linear input/output relationship of the SAMs (*Shomrat et al., 2011*). Since the activity-dependent changes at the SFL-SAM synapse involve pre- and postsynaptic mechanisms (see below), we suggest that, rather than being simple relay interneurons, the SAMs provide a synaptic 'gain-control relay' of the SFL-LB input into the CAMs and LNs.

## The CAMs as a putative feedforward inhibitory neural network balancing the parallel SAM feedforward excitatory neural network

The newly discovered CAMs comprise only 1.6% of the VL cortex. An inhibitory nature of CAMs is supported by GABA immunolabeling of putative CAM dendritic processes at the level of the SFL tract and in varicosities in the neuropil (Stern-Mentch et al., 2022). Also, inhibitory CAMs would explain the IPSPs or EPSP followed with a short delay by an IPSP recorded in some of the LNs in response to SFL tract stimulation (*Figure 7—figure supplement 1*).

Although the CAMs show morphological diversity (*Figure 6—figure supplements 2 and 3*), they share similar connectivity, integrating multiple inputs from the SFL axons, SAMs, and CAM-specific inputs (CIN) in their bifurcating dendrites (*Figure 6B*). The CAMs innervate the LN dendritic processes only sparsely, unlike the dense LN innervation by the SAMs (*Figure 7B and D*). This innervation pattern suggests that the LNs integrate stimulus-specific excitatory inputs from many SAMs, while the CAMs, which integrate ubiquitous and nonselective inputs, are optimally located for inhibitory balancing the overall SAMs excitatory inputs.

If the CAMs are indeed inhibitory interneurons, then our results strongly suggest the VL network is organized into two interconnected parallel feedforward networks, in which the SFL axons convey sensory inputs that 'fans-out' onto SAMs (in ~1/12 ratio) and 'fan-in' (~4/1) onto the CAMs. Both the SAMs and CAMs feedforward onto the LNs in the VL readout layer (schematically shown in *Figure 10A*). This type of organization may sharpen the stimulus-specific SAM inputs to the LNs by subtracting the global or background activity monitored by the inhibitory CAMs. This type of organization may highlight a universal inhibitory balancing organization in networks with different functions. A similar organization is found in cortical circuits representing sparse, odor-specific, activity where the

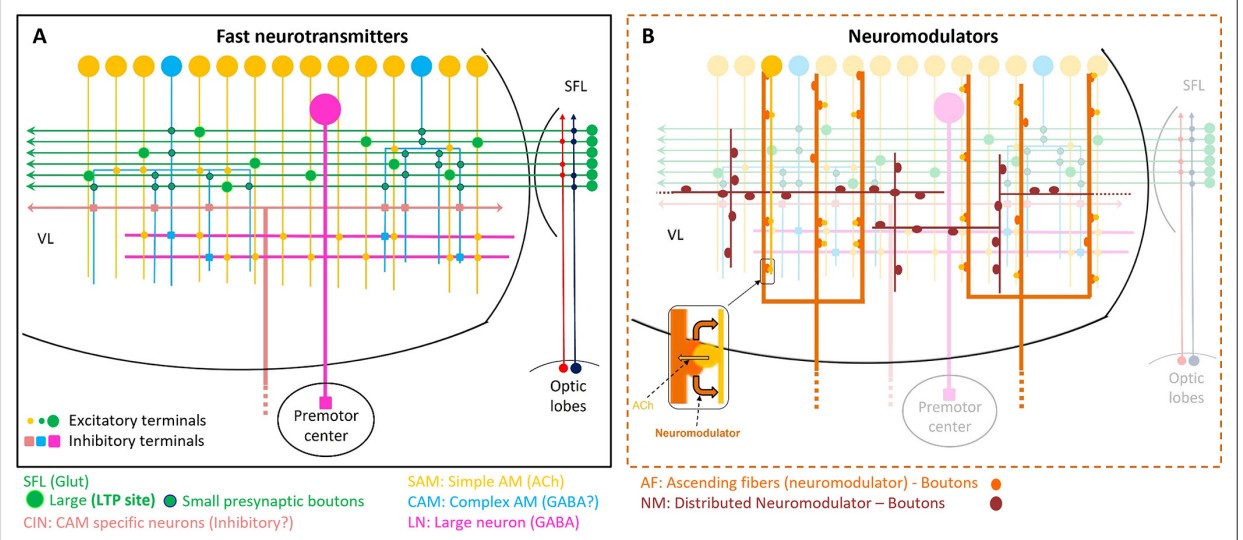

**Figure 10.** Schematic representation of the currently known circuit architecture of the vertical lobe (VL). (**A**) Fast transmitter connectivity. The parallel superior frontal lobe (SFL) axons (green) convey classified sensory information via *en passant* glutamatergic (glut) synapses to a large group of simple AMs (SAMs) (yellow) and a very small group of complex AMs (CAMs) (cyan). Each SAM has one palm on its neurite innervated by a single large SFL bouton (SFL LB). This connection is endowed with short- and long-term plasticity. The CAMs integrate ongoing activity through multiple inputs from the SFL, SAMs, and CINs (light pink). The SAMs and CAMs converge onto the large efferent neuron (LN) dendrites (magenta) forming two parallel and interconnected feedforward networks (shown in *Figure 11*). SAM output to the LNs is excitatory cholinergic (ACh). The CAM inputs to the LNs may be inhibitory GABAergic (see text). (**B**) The two widespread networks of modulatory inputs to the fast transmission circuit (shown in A). The ascending AF processes (orange) make reciprocal connections with the SAM neurites, (shown in the zoom-in inset) possibly inducing the local release of a neuromodulator (likely serotonin). The neuromodulatory fibers (NMs) (brown) spread throughout the neuropil. Physiological and immunohistochemical studies indicate that ascending fibers (AFs) and NMs release neuromodulators such as serotonin and octopamine and dopamine that likely play a role in reinforcing/suppressing the LTP at the SFL-to-SAM synapse.

sparse activity is generated through summing unbalanced widespread and broadly tuned inhibition with odor-specific excitation (*Poo and Isaacson, 2009*). In the octopus VL, such a sharpening mechanism may ensure recognition of learned visual scenarios even under variable light intensities. Theoretical studies have demonstrated the importance of implementing feedforward inhibition parallel to the feedforward excitation to balance the stable propagation of an activity wave (*Litvak et al., 2003*). Feedforward inhibition also appears important for balancing networks' input/output relationships to avoid attraction of network activity into either up or down states (*Ferrante et al., 2009*). Finally, longterm balancing inhibition is believed to be important in ensuring network homeostasis, especially in networks involved in learning and memory, like the hippocampus, where activity balances may be perturbated by long-term activity-dependent plasticity (*Turrigiano and Nelson, 2004*).

## A reciprocal interaction between the SAMs and the AF may provide a mechanism mediating a 'self-LTP' induction

That the SAMs make synaptic contact with the neuromodulatory boutons of AF-Nm and AF-mix fibers (*Figure 9*) raises the possibility that the AF system sends both extrinsic and intrinsic neuromodulation signals to the VL network, like, for example, the neuromodulatory control of the rhythmic output of central pattern generators (CPGs) (*Nusbaum et al., 1992*; *Katz et al., 1994*; *Katz and Frost, 1996*). The AF-Nm and AF-mix may transmit extrinsic 'teaching' neuromodulatory signals from the Sub-VL into the VL (see below). The SAM synapses on the boutons and processes of the two AF systems (*Figure 9A–D*) strongly suggest that SAMs may also evoke the local extrasynaptic release of neuromodulators. This circuit motif, schematized in *Figure 10B inset*, suggests a mechanism for mediation of activity-dependent local release neuromodulation, which in turn heterosynaptically mediates synaptic plasticity of the adjacent neuronal elements.

This hypothesis seems particularly relevant for the interaction of the SAMs with the AF-Nm system, especially if our identification of the AF-Nm as serotonergic is correct. The intense SAM-to-AF-Nm connection (*Figure 9B*) and the rosette-like structure in which several SAMs make adjacent synapses

onto one AF-Nm bouton, mostly filled with dense-core vesicles, suggests a reciprocal interaction between excitatory interneurons (SAMs) and neuromodulatory processes (AF-Nm). The premise that AF-Nm is the source of serotonergic signals is supported by *Shigeno and Ragsdale, 2015* finding of serotonergic immunoreactivity in processes climbing into the cell body cortex in the lateral lobule, strikingly similar to the distribution of AF-Nm processes described here. This arrangement raises the intriguing possibility that during LTP induction the strongly activated SAMs evoke the local release of 5-HT from the AF-Nm bouton, which, in turn, through extrasynaptic transmission, activates metabotropic 5-HT receptors in the SAM. The coincidence of 5-HT with the activity-dependent mechanism in the SAM, could trigger a biochemical cascade that activates NOS only in the SAMs strongly activated by their SFL inputs. The finding that 5-HT induces short-term synaptic facilitation and reinforces the activity-dependent LTP induction (*Shomrat et al., 2010*), supports the feasibility of a 5-HT-mediated 'self-LTP' induction mechanism.

The suggested 5-HT involvement in the modulation of transmitter release in the VL resembles the heterosynaptic 5-HT-induced facilitation of transmitter release in the defensive reflex of the gastropods mollusk *Aplysia* (*Hochner et al., 1986*; *Ghirardi et al., 1992*; *Bailey et al., 2000*) and points to an evolutionary conservation of molluscan serotonergic neuromodulation mechanisms. We suggest that, like the mechanosensory neurons of *Aplysia* (*Hochner and Glanzman, 2016*), the SAM interneurons activate the LNs and in parallel also evoke 5-HT release from the AF-Nm processes which, in turn, facilitate the same SAMs input to the LNs.

## How can *synaptic specificity* and *associativity* be achieved with a single SFL-to-SAM input?

*Specificity* and *associativity* are essential properties of associative learning networks. In this context, synaptic *specificity* means that synapses not active during learning are not strengthened, while synaptic *associativity* ensures the strengthening of only the synapses that together produce a strong enough postsynaptic response during learning. The textbook model for these Hebbian LTP properties is the input of Schaffer collaterals to the pyramidal cell of the CA1 region of the hippocampus. In this synapse, in contrast with the VL, activity-dependent LTP *induction,* and synaptic *specificity* and *associativity* are mediated by a postsynaptic mechanism activated by a strong enough postsynaptic response that opens an incident-detecting glutamate-gated NMDA receptor channels (*Malenka, 2003*; *Kandel et al., 2012*).

To understand how synaptic *specificity* could be achieved within the unique wiring of the VL we note that LTP expression in the VL involves an NO-dependent presynaptic facilitation of transmitter release from the SFL boutons. It was suggested that NO is generated by NOS which is persistently activated in the SAMs following activity-dependent LTP induction (*Turchetti-Maia et al., 2018*). Accordingly, to ensure synaptic *specificity*, NO diffuses retrogradely from the postsynaptic (SAM) into the presynaptic terminal (SFL LB) where it induces NO-dependent presynaptic facilitation of transmitter release. The characteristic morphological arrangement described here, with the SAM palm partly wrapping the presynaptic SFL LB, can ensure synaptic *specificity* as the steep NO concentration gradient will affect only these closely attached presynaptic boutons (*Hall and Garthwaite, 2009*).

What are the possible mechanisms of LTP *associativity*? LTP maintenance can be mediated through a molecular switch mechanism consisting of a positive feedback loop of NO-dependent NOS persistent activation (see details in *Turchetti-Maia et al., 2018*). This molecular switch provides a tentative molecular model for associativity, whereby volume-type transmission of NO (*Garthwaite, 2016*) between the adjacent activated SAMs neurites may mutually facilitate the NO-dependent switching on of NOS in those SAMs that were strongly activated together. As we discuss below, many SAMs together with a few CAMs appear to be organized in what we term an 'association module' as the SAMs (and CAMs) neurites converge within the neuropil to form compact bundles of thin intermingling processes traveling together (*Figure 7A*). Such spatially closed arrangements of thin neurites (e.g. *Figure 7A and C*), which express intense NOS activity (*Turchetti-Maia et al., 2018*; Stern-Mentch et al., 2022), can facilitate the mutual buildup of NO concentration in the volume shared by the adjacent SAM neurites.

## Perspective on the VL involvement in Octopus associative learning

In contrast to the canonical hippocampal Hebbian type associative LTP, which is mediated postsynaptically by the NMDA-receptor (*Malenka and Nicoll, 1999*), here, we suggest an alternative mechanism

where the association mechanism depends on the local accumulation of NO (see above). Yet, in the VL, as well, the coactivated SAMs collectively evoke a local postsynaptic response in the LNs dendritic processes, while presynaptically producing a summated level of NO around the SAMs neurites. Thus, the local postsynaptic response in the inexcitable LNs dendrites (*Shomrat et al., 2011*; *Figure 7—figure supplement 1*) is likely to be correlated with the local presynaptic NO concentration - creating a correlation between the magnitude of the presynaptic SAMs coactivity (NO) (which mediate synaptic associativity (see above)) and the postsynaptic responses (EPSP) - a relationship important for establishing a Hebbian-type synaptic associativity (*Malenka, 2003*). Such organization possibly represents an adaptation of a synaptic association mechanism for a network based on a feedforward connectivity.

## Perspective on the involvement of the AF and the NM in the supervision of learning

Octopuses quickly learn to discriminate between positively and negatively rewarded targets in various forms of conditioning paradigms (see review by *Zarrella et al., 2015*). These forms of learning, particularly passive avoidance learning, were shown to depend on the integrity of the SFL input to the VL and the capacity of the VL to undergo LTP (*Shomrat et al., 2008*). Training on these discriminative tasks involves the activation of a supervising 'teaching' signal in the form of a reward (such as food) for a correct choice or a punishment (such as an electric shock) for an incorrect choice. These teaching signals are likely transmitted into the VL neuropil via action potentials propagating along the neuromodulatory processes. Some of their cell bodies, like those of serotonergic cells, are located in the sub-VL (*Shomrat et al., 2010*; *Shigeno and Ragsdale, 2015*) suggesting this is the source for the global 5-HT-dependent plasticity regulation in the VL. Since we have described two ascending fiber systems (AF-Nm and AF-Mix), it is tempting to speculate that they may fulfill different functions. The AF-Nm processes, which interact more intensely with SAMs than the AF-mix, maybe mostly involved in mediating the intrinsic SAMs-induced local extrasynaptic release of the AF-Nm neuromodulator (most likely 5-HT). The AF-Mix, on the other hand, with its occasional synaptic outputs to the SAMs and boutons filled with a mix of smaller vesicles, may better fit the role of transmitting a global supervision signal that comes from outside the VL, where propagating action potentials evoke global neuromodulators release.

The idea that the AF fibers transmit a global serotonergic teaching signal fits well with the context-dependent flexibility of octopus attack behavior. We propose that during training on an avoidance task, the AF systems transmit a serotonergic 'punishment' signal into the VL. This 5-HT signal, evoked by a noxious stimulus to the arms, facilitates the induction of a long-term association between the visual features that characterize the target which was negatively rewarded (e.g. brightness and shape) (*Shomrat et al., 2010*). As a result, after training (which in *Octopus vulgaris* will take about ten trials; *Shomrat et al., 2008*) the 'trained' group of SAMs will 'remember' to respond strongly to the 'memorized' features, resulting in a stronger activation of LNs in the VL output layer. As the LNs are GABAergic (Stern‑Mentch et al., 2022) the memorized association will produce a sufficiently strong VL inhibitory output to override the reflexive tendency of the octopus to initiate its stereotypic attack behavior (see details *Shomrat et al., 2008* tentative octopus L&M model in *Turchetti-Maia et al., 2017*).

The rich and widely distributed neuromodulatory processes (NMs) intermingle with the fast transmitter feedforward network (schematized in *Figure 10B*). The fibers form a mesh in the neuropil and may be associated with the regulation of overall VL activity. Their exceptionally large boutons, frequently lacking typical release sites, are filled with a variety of large dense-core vesicles of the type usually storing and releasing neuropeptides and non-classical neurotransmitters (*Zhu et al., 1986*; *Leng and Ludwig, 2008*; *van den Pol, 2012*; *Witvliet et al., 2021*). A variety of neuropeptides and hormones have been identified in the *Octopus vulgaris* VL - serotonin, dopamine, octopamine, FMRFamide-related peptides, vasopressin, oxytocin, octopressin, Y-peptide and more (*Shomrat et al., 2010*; *Shigeno and Ragsdale, 2015*; *Winters et al., 2020*; Stern‑Mentch et al., 2022). Especially interesting is that tyrosine hydroxylase (TH) - labeled processes show a wide distribution in the VL neuropil (Stern‑Mentch et al., 2022), similar to that of the NM processes (schematized in *Figure 10B*), suggestive of a catecholaminergic neuromodulatory innervation, most likely dopaminergic. Uniquely for the catecholaminergic system, a group of large TH-positive cell bodies is organized in what appears to be a 'deep nucleus' located at the border between the SFL and VL. This suggests that, in contrast to the

external source of the serotonergic signals of the AF, dopamine may convey internal neuromodulatory signals within the SFL-VL system (see Stern-Mentch et al., 2022). Unpublished results suggest that dopamine has a short-term facilitatory effect, like 5-HT, but in contrast to 5-HT, dopamine suppresses the transition to the long-term maintenance phase of LTP.

## What is the role of the CAM-specific input fibers (CIN)?

The exclusive innervation of the very small group of CAMs by a distinct group of processes (CIN) must be functionally significant (schematically shown in *Figure 10A*). The CIN processes resemble fibers that Young speculated were ascending fibers carrying 'pain signals' from the arms and mantle (*Gray, 1970*; *Gray and Young, 1964*). Unlike the NMs and AFs that barely display any presynaptic profiles, the CIN outputs to the CAMs have the typical morphological features of fast transmitter synapses (*Figure 8*). The presynaptic features are similar to vertebrate presynaptic terminals with small electron-dense synaptic vesicles, which are occasionally associated with inhibitory synapses (*McDonald et al., 2002*; *Kasthuri et al., 2015*). We, therefore, speculate that in addition to feedforward excitation from the SFL axons and SAMs, the CAMs integrate inhibitory inputs from the CIN. The CINs may provide feedback adjusting the level of LNs output to that of its targets in the subvertical lobe to where the LNs axons project, and which Young described as the origin of the 'pain fibers.' Such inhibitory feedback may 'focus' the VL output by means of global up- and down-regulation of the balancing inhibition conveyed by the CAMs.

In the framework of the VL model (*Shomrat et al., 2015*; *Turchetti-Maia et al., 2017*), in which the GABAergic LN output (Stern-Mentch et al., 2022) inhibits the octopus' tendency to attack (*Shomrat et al., 2008*), the CIN inhibitory input onto the CAMs would lead to an increase in the LN output as a result of the reduction in the inhibitory drive from the CAMs. In 'emergencies' a strong 'panic' activation of the CIN would lead to an acute unconditioned suppression of the attack behavior via shutting down the CAMs inhibitory inputs to the LNs.

## Many SAMs together with a few CAMs appear to be organized in a canonical 'association module'

Our reconstructions highlighted that ensembles of closely intermingled groups of SAM and CAM neurites form a *quasi*-columnar structural unit (*Figures 7A, C, 9A and E*, *Figure 9—figure supplement 1*). This group of closely packed thin neurites possibly functions as an 'association module.' The neurons in each module may respond to a combinatorial ensemble of sensory features from the SFL, as schematically represented in *Figure 11* **inset**. Such combinatorial coding is a common feature in association networks such as the insect and mammalian olfactory systems (*Malnic et al., 1999*; *Lledo et al., 2005*; *Galizia and Szyszka, 2008*; *Turner et al., 2008*; *Gruntman and Turner, 2013*; *Kurian et al., 2021*). However, the one-to-one SFL-to-SAM connections suggest that each SFL neuron is represented in $12 \pm 3.6$ SD SAMs, spread randomly over the VL cortex (*Figure 11*). This suggests that each of VL association module associates many (i.e. several hundreds of SAMs) random combinations of visual features that are represented in the $1.8 \times 10^6$ SFL neurons. In principle, the 'association module' is equivalent to that of a single interneuron in the input layer of a classical 'fan-out fan-in' classification network as proposed in *Shomrat et al., 2011*. It is tempting to speculate that what is achieved in other systems by a single complex neuron is achieved in the octopus by the construction of a microcircuit composed of many neurons with simple properties (e.g. a linear input/output relationship).

## How well does a partial connectome of a small portion of one VL lateral lobule represent the connectivity patterns across all five VL lobuli?

One limitation of our dataset, as in most of the large tissue connectome studies, is that the ROI ($260 \times 390 \times 27$ μm) constitutes only a very small part of the vertical lobe (~$3 \times 3 \times 4$ mm). Accordingly, the question arises whether the neuronal network described here can be generalized to the five VL lobuli. Several lines of evidence suggest that the generalization is highly plausible for the connectome of the fast transmission circuitry (i.e. SFL-to-SAM, SFL-to-CAM, SAM-to-CAM, SAM-to-LN, CAM-to-LN, CIN-to-CAM). First, these synaptic motifs were widely observed in the small dataset from a middle VL lobule of the same animal. Second, morphological, histological, and immunobiological analysis have

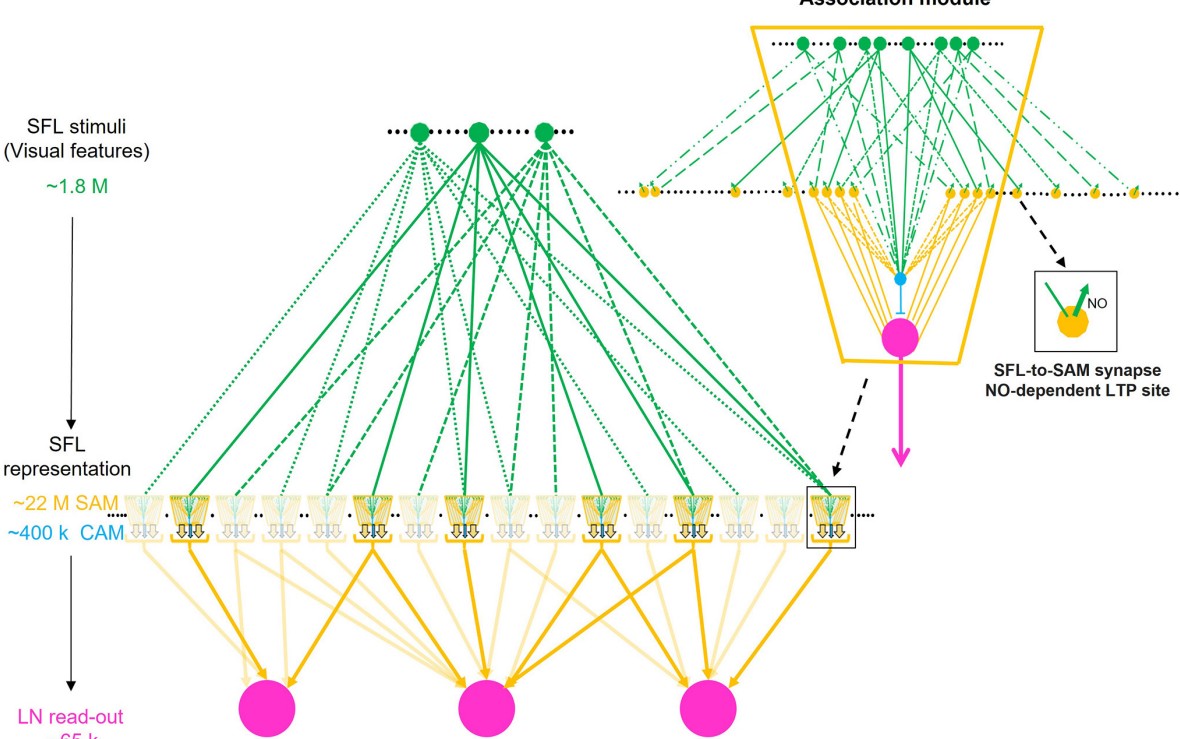

**Figure 11.** Schema of combinatorial coding in an array of possible association modules. *Inset:* The simple AMs (SAMs) and complex AMs (CAMs) appear to be organized in columnar structures that integrate and feedforward 'sharpened' stimulus-specific inputs into the large efferent neurons (LNs). A specific stimulus (e.g. a visual feature) is fed forward via the SAMs while the CAMs integrate multiple superior frontal lobes (SFLs) and SAMs inputs and feed a general balancing inhibitory input forward to the LNs. As the input to the SAMs is endowed with long-term potentiation (LTP) (zoom-in inset), this microcircuit unit may be a canonical 'association module' of the vertical lobe (VL). *The main scheme* shows how an array of association modules integrates a very sparse and random distribution of the SFL single input to the SAMs. Each association module likely contains several hundred SAMs and relatively few CAMs (1.8%) suggesting that each module associates a variable combinatorial association of multiple SFL features. Note that computationally an association module may replace a single AM in the intermediate layer of the originally assumed fan-out fan-in classification/association network (*Shomrat et al., 2011*).

revealed shared features across the five lobuli (*Gray and Young, 1964*; *Gray, 1970*; Stern-Mentch et al., 2022) and by and largely present a grossly similar distribution of the main neurotransmitters (GABA, glutamate, and acetylcholine) across lobuli (Stern-Mentch et al., 2022). Third, physiological studies have shown that the principal synaptic connectivity properties are shared by all VL lobuli. The glutamatergic connection of the SFL-to-AM, the site of LTP, and the cholinergic AMs-to-LN connections demonstrate similar properties in all five lobuli (*Hochner et al., 2003*; *Shomrat et al., 2011*). We, therefore, anticipate that the fundamental circuit motifs of the fast transmission system identified here are not unique to our ROI but instead represent a general feature of VL connectivity. In contrast, neuromodulators, neurohormones, and neuropeptides have distinct neurochemical signatures across the VL lobuli (*Shigeno and Ragsdale, 2015*; Stern-Mentch et al., 2022), suggesting the existence of anatomical compartmentalization. This is further supported by notable differences in the involvement of NO in LTP (*Turchetti-Maia et al., 2018*) and the differential effects of octopamine and dopamine on LTP maintenance across VL lobuli (unpublished results).

Accordingly, we hypothesize that different functional compartmentalization could be established in different areas of the VL through the differential distribution of neuromodulation systems superimposed on a hard-wired connectivity network (*Bargmann, 2012*).

In summary, while earlier reports (*Gray and Young, 1964*; *Gray, 1970*; *Young, 1971*) suggested that the VL is organized in a simple fan-out fan-in feedforward architecture much like an associative 'perceptron' network (*Shomrat et al., 2011*), our study has revealed that its organization is more complex. The VL circuitry is characterized by two parallel and interconnected feedforward networks

assembled in what appear to be association modules, in which the SFL axons conveying processed sensory inputs fan-out onto many SAMs and fan-in onto a low number of CAMs. The networks originating from the excitatory SAMs and the probably inhibitory CAMs feedforward stimulus-specific excitatory and global balancing inhibitory inputs into the LNs of the VL output layer.

## Methods

### Tissue preparation

A mature wild *Octopus vulgaris* (body weight~130 g) was collected by a local fisherman on the Israeli Mediterranean coast and was held in artificial seawater in an individual aquarium for a few days. As previously published (*Shomrat et al., 2011*), the animal was deeply anesthetized in seawater containing 55 mM MgCl$_2$ and 2% ethanol. The supraoesophageal mass was removed through a dorsal opening of the cranium (*Young, 1971*) and then quickly sliced into 1 mm sagittal sections using a mouse matrix brain slicer and the double-edged razor blade. Three slices containing the vertical lobes were obtained and immersed in a fixative solution (20:1 ratio).

### Fixation

Because preservation of the ultrastructure is a central requirement of a connectomics study, three fixation protocols were tested, chosen for their various aldehyde concentrations and tonicity, as well as their previous use with octopus or other marine invertebrate tissue preparations for electron microscopy. Each slice was immersed in one of the following:

Fixative #1 (*Kier, 1996*): 3% glutaraldehyde, 0.065 M phosphate buffer (PB), 0.5% tannic acid, and 6% sucrose for 12 hr at 4 °C.

Fixative #2 (*Feinstein et al., 2011*): a mixture of 3% glutaraldehyde and 4% paraformaldehyde formaldehyde in artificial seawater for 4 hr at room temperature (RT).

Fixative #3 (*Schmidt and Derby, 2011*): 5% glutaraldehyde in 0.1 M PB containing 15% sucrose for 4 hr at RT.

After fixation, the slices were rinsed in 0.1 M PB for 4×30 min, then transferred to chilled 0.065 M phosphate buffer with 1% glutaraldehyde and 6% sucrose and stored at 4°.

We chose fixative #1 for the volumetric reconstruction, constituting the 'large connectomics dataset' (4 nm/pixel, 200 ns/pixel), due to good preservation of the macroelements (i.e. cell body) and ultrastructure (e.g. mitochondria, postsynaptic density, various types of synaptic vesicles, etc.) as well as the high contrast between elements. Fixative #3 showed poor preservation of the macroelements but greater expansion (more extracellular space) and overall better fine ultrastructural preservation. Accordingly, this fixative was used to generate a small 'high-magnification dataset' (1 nm/pixel, 2 μs/pixel) to examine possible synaptic specialization and the synaptic vesicles at the sites of LTP generation in the VL (*Hochner et al., 2003*; *Shomrat et al., 2011*).

### EM staining

The samples were stained with a modified *en bloc* staining protocol developed by *Hua et al., 2015* for volumetric EM imaging. Before staining, the chemically fixed tissues were rinsed three times for 20 min each in 150 mM sodium cacodylate buffer (pH 7.4). The tissues were then stained in 2% w/v OsO4 buffered with 150 mM cacodylate for 2 hr. After rinsing 3 × 20 min in cacodylate buffer the tissues were immersed in 1.5% w/v potassium ferrocyanide buffered with cacodylate for 4 hr. The tissues were rinsed in deionized water for 3 × 20 min and then incubated in filtered 0.5% w/v thiocarbohydrazide aqueous solution for 1.5 hr. The tissues were then rinsed in deionized water for 3 × 20 min and stained again with 2% w/v OsO$_4$ aqueous solution for 3 hr. After the second osmication, the tissues were rinsed in water for 3 × 20 min and transferred to 1% w/v uranyl acetate aqueous solution which was wrapped in aluminum foil to protect from light for overnight staining (approx. 12 hr, RT). On the next day, the samples were rinsed in water and then dehydrated through 25%, 50%, 75%, 100%, and 100% acetonitrile solution for 20 min each into pure acetonitrile. The dehydrated samples were infiltrated with 25% EPON:acetonitrile for 6 hr, 50% EPON:acetonitrile for 12 hr, 75% EPON:acetonitrile for 12 hr, 100% EPON for 12 hr, 100% EPON for another 12 hr. The fully infiltrated samples were placed in a flat embedding mold and incubated in a 60 °C oven for two days.

## Collection of serial ultrathin sections using an automated tape-collecting ultramicrotome (ATUM)

The block face of the sample cylinder was trimmed into a 1 × 1 mm square containing the vertical lobe. 30 nm thin sections for the 'connectomics dataset' and 40 nm sections for the 'high magnification dataset' were cut using a Leica ultramicrotome and collected on a home-built automated tape collecting system using a plasma-treated polyamide 8 mm wide tape (Kapton, Sheldahl) that was carbon-coated (*Hayworth et al., 2014*). On collection of the sections, the tape was cut into strips, which were mounted on 100 mm silicon wafers. The connectomics dataset was comprised seven wafers, between 115–140 sections per wafer, (total of 892 sections), while the high magnification dataset consisted of a single wafer of 83 sections.

## EM imaging

For the connectomics dataset, the sections were first mapped at low resolution (*Figure 2A*) and the EM image stack was aligned by Fiji's plugin 'Linear Stack Alignment with SIFT' to form a coherent 3D EM volume. The VL region free from large cracks in the tissue was chosen for high-resolution imaging. The sections on the wafer were cleaned while in the Magellan microscope using the FEI-supplied plasma system. An FEI Magellan Scanning Electron Microscope (Thermo Fisher Scientific) equipped with custom image acquisition software (WaferMapper, *Hayworth et al., 2014*) was used for high-resolution imaging. A backscattered electron detector in long working distance mode with a 3.2 nA beam current at 7 kV incident electron energy and a dwell time of 200 ns/pixel was used for imaging at 4 nm/pixel. For each section, we obtained a 4 by 6 montage of image tiles (each tile was a 16,000 × 16,000-pixel image). At the end of each imaging run, out-of-focus images were manually identified and retaken. Artifacts such as dirt, lead precipitates, wrinkles, or even large folds were seen on many sections and severely damaged sections were removed from the total stack. This resulted in 51 sections missing from the 892 sections. However, the occurrence of serious damage in adjacent sections was rare, allowing confident tracing of the elements.

## Image stitching and alignment

Stitching and alignment of the dataset were performed using a method inspired by *Saalfeld et al., 2012*. Each step was parallelized over image tiles and sections. The single-beam electron microscope produced high-resolution image tiles sized 16,000 × 16,000 pixels whose precise location relative to one another was not known at the nanometer level. Hence, stitching these tiles together required image information. We stitched these image tiles for each of the 892 sections using first a coarse-grained stitching followed by fine-grained stitching. The coarse-grained stitching performed a per-tile affine transformation, setting the tiles roughly in their final position. SIFT features (*Lowe, 1999*) were computed for pixels up to 2,400 from the tile boundary. The features of the overlapping tiles were compared and matched to find point correspondences., the RANSAC algorithm (*Fischler and Bolles, 1981*) removed outliers of incorrect correspondences, optimizing for rigid transformations. Finally, the tiles' affine transformation was optimized by minimizing the sum of the squared roots of all correspondences.

The fine-grained stitching achieved better alignment and compensated for minor non-linear deformations in imaging. A triangular mesh was laid on each tile and a 1 × 1 µm2 area around each mesh vertex overlapping with an adjacent tile was cropped. The cropped patch was matched against a transformed cropped area of the neighboring tile of size 2 × 2 µm2. The 2D elastic optimization algorithm described in *Saalfeld et al., 2012* was used to minimize the distances of the matches.

We applied a custom-made 3D alignment algorithm to obtain an image stack of all 892 sections. Once all the sections were stitched as described above, correspondences between adjacent sections were detected. The first step consisted of coarse feature detection and matching, and subsequent patch matching in small regions. The matching was performed between all section pairs separated by at most one section (hence considering matches between non-adjacent sections). Coarse features were obtained from the stitched sections and were detected using the OpenCV SimpleBlobDetector ('OpenCV: cv:SimpleBlobDetector Class Reference' n.d.). The detector searched for circular blobs that typically corresponded to blood cells, mitochondria, and axonal cross-sections. Features were defined using a SIFT descriptor (*Lowe, 1999*). After matching the features of each pair of sections, RANSAC filtered out wrong matches, from which an affine transformation between the sections was

computed. The second step was fine-grained patch matching at half resolution. A triangular mesh was laid on the sections and a 1.6 × 1.6 μm² area around each mesh vertex was cropped, followed by a cross-correlation search against a transformed cropped area of the neighboring section of size 4 × 4 μm2. The valid cross-correlation matches were then used as an input to a 3D elastic optimization algorithm based on *Saalfeld et al., 2012*, which minimizes the cross-layer distance of the matches while preserving the 2D structure of the sections.

## Semi-automatic segmentation: mEMbrain

To segment the aligned image stack (892 sections, volume 2.7 million μm³; *Figure 1B*), we applied an automated reconstruction pipeline to the volume using a machine learning reconstruction pipeline (*Lee et al., 2017*; *Meirovitch et al., 2019*). The inputs to this pipeline required fully annotating the image stacks of four manually picked volumes (image stacks ranged in volume from 21.6 to 153 μm³; *Figure 1—figure supplement 2*), which served as ground truth for training the artificial neural network. The annotation of this ground truth labeled two different categories: intracellular space, excluding cellular membranes (category 1), or any combination of cellular membrane and extracellular space (category 2). We leveraged the mEMbrain software package implemented in MATLAB (*Pavarino et al., 2023*) to obtain a first template of the cellular processes. This computation included training a deep neural network (Unet; *Ronneberger et al., 2015*) to classify pixels according to the two categories (accuracy above 93%) and produce a 2-dimensional instance segmentation of the individual cross-sections of all cellular compartments using Watersheds (*Pavarino et al., 2023*).

We used custom code to merge colored neuronal cross-sections across slices of the image stack if adjacent cross-sections matched in shape or if adjacent processes had sufficient overlapping cytoplasm far from any cellular membrane. Applying this conservative merging resulted in a 3-dimensional representation of object instances that each rarely overlapped two distinct neuronal or glial processes and also often the splitting of complete neuronal processes into several segmented objects. In most cases, however, the 2-D cross-sections were lawfully (fully) segmented into one object, and after a conservative agglomeration procedure based on shape matching these objects were correctly agglomerated over a few consecutive sections. We also applied 3D agglomeration procedures including mean affinity agglomeration suggested by *Lee et al., 2017* and *Meirovitch et al., 2019*. However, due to membrane breaks these procedures often resulted in merge errors that were hard to manually proofread, hence, this agglomeration was not included in the circuit analysis. We found that the merge errors were associated with specific cellular and compartment types, including the shafts of the large neurons and many synaptic boutons. Because such preparation artifacts severely affected the quality of our fully automated reconstruction, we combined the automated segmentation with the manual methods described below.

## Membrane-constrained ML-aided reconstruction

To facilitate manual reconstruction we used the mEMbrain package (*Pavarino et al., 2023*) implemented in MATLAB (Mathworks) together with VAST Volume Annotation and Segmentation Tool (https://software.rc.fas.harvard.edu/lichtman/vast/, *Berger et al., 2018*). Using U-net architectures of deep neural networks (*Ronneberger et al., 2015*) mEMbrain computed the boundaries of all cellular processes, scoring the likelihood (0–1) that pixels belonged to intracellular space (category 1) or not (category 2). mEMbrain arranged these scores into a volume of membrane probabilities consistent with the EM image stack layer in VAST, which allowed tracing cellular processes aided by the pre-computed membrane probabilities. We constrained the tracing to the membrane probabilities layer generated by mEMbrain using the 'constrained painting' function in VAST, which restricted the tracing to connected components of pixels far from cellular membranes. This ML-aided tracing allowed the annotator to ignore the location of the cellular membranes and work in a skeletonization mode if the initial painting stroke overlapped intracellular space. Tests on small volumes suggested a~5–10 x faster annotation in this mode than manual painting without constraints, while retaining the accuracy of careful pixel annotation. This membrane-constrained ML-aided reconstruction was used for the 516 cells described and studied morphologically here (*Figure 1C*).

## Node-based agglomeration of 3D compartments

We also converted the automated neuron reconstruction IDs into a VAST image stack layer holding the identities of ~16 × 10⁶ objects to allow even faster semi-automated reconstruction of processes. In this procedure, the annotator used the automatic reconstruction by iteratively agglomerating 3D objects that were found to belong to a single cellular process. This was done using VAST's 'Tool Layer' functionality which we mapped to the automated segmentation layer and to an additional annotation layer that represented the annotator's manually made skeletons. The annotator placed single nodes within the objects segmented by the automated reconstruction pipeline; together the collection of nodes of a single skeleton defined the agglomeration of all objects straddled by the nodes into a single proofread cell. This membrane-constrained ML-aided reconstruction was used to quickly reconstruct all the synaptic partners of one LN (*Figure 7C*) and one AF (*Figure 9—figure supplement 1*).

## Criteria for cell classification

Our morphological reconstruction included segments of 516 cells and a wiring length of more than 12.8 cm. Only neurons that were sufficiently contained within the reconstructed volume were classified. Processes that could not be followed across the volume or were not wholly contained within it were considered orphan fragments and fell into the 'unmarked category' with other uncharacterized profiles.

### Afferent axons

Classified as afferent axons were processes of constant radius (as opposed to dendrites that often taper), with exclusively presynaptic boutons filled with synaptic vesicles <60 nm and with no connection to a cell body. Two processes met these features: the SFL axons see Results; *Figure 3* and the CIN (see Results; *Figure 8*). These were classified according to several established criteria (*Gray and Young, 1964*; *Gray, 1970*; *Young, 1971*): the axon projected into the VL lobule within the outer zone of the neuropil, made *en passant* synapses with AM neurons and at the ultrastructural level their varicosities showed an electron-lucent cytoplasm with average synaptic vesicle size of 59.7 ± 1.5 nm, and numerous round mitochondria. Unlike the SFL, CIN was found in the SFL and in the neuropil. These displayed the smallest vesicles in our dataset (38.2 ± 0.9 nm) and synapsed exclusively with clear postsynaptic profiles. Additionally, CIN fibers showed a high concentration of neurofilaments which ended where the synaptic vesicles appeared (*Gray, 1970*). The high neurofilament content of these processes led *Gray, 1970* to suggest that these were pain fibers. Both afferents could be easily identified with only electron micrographs.

### VL interneurons

Neurons were classified as VL interneurons if they received input from the SFL and/or were previously described as intrinsic to the lobes. All partners of the SFL axons were the AM cells (*Gray and Young, 1964*; *Gray, 1970*; *Young, 1971*). However, we discovered two types of AMs: SAM and CAM (see Results; *Figure 6*). The SAMs showed a single inwardly directed trunk that cannot be defined either as an axon or dendrite. Their ultrastructural signature was easily identifiable through the presence of synaptic vesicles (49.5 ± 1.3 nm) along the length of this trunk and the presence of specific ultrastructural components such as cisternae and the transport endoplasmic reticulum at every presynaptic site. Unlike the SAMs, the CAMs displayed a strict segregation between a dendritic compartment located within the SFL tract and their axonal process within the neuropil. Their dendritic tree was lightly or heavily branched, with one or multiple axons arising from it. Their dendritic arborizations displayed a classical electron-lucent postsynaptic profile filled with polyribosomes. Unlike the SAMs, the CAM presynaptic sites in the neuropil displayed a clearer cytoplasm with no specific ultrastructure and smaller vesicles than the SAMs (42.6 ± 1.1 nm).

We also observed putative interneurons displaying a bifunctional neurite, with both pre- and postsynaptic sites identified (*Figure 1—figure supplement 1A*). Previously undescribed and poorly represented in our dataset and apparently poorly involved in the major cell network, all such cells were considered as unmarked cells in the data analysis.

### Efferent fibers

Only the LNs fell into this category. Their cell bodies (diameter >15 µm; n=14) lay mostly in the inner zones of the cortex and their large trunks ran radially into the neuropil (see Results; *Figure 7*). While most of the LNs were poorly contained in the reconstructed volume, some gave off dendritic collaterals within the SFL tract area where they were solely postsynaptic to the SAMs and unmarked process. Their dendrites showed a variable diameter along their length and displayed a typical clear postsynaptic profile. In the neuropil, we classified all clear postsynaptic profiles with convoluted arborizations heavily innervated by the AMs as LN processes.

### Neuromodulatory neurons

Based on known invertebrate modulatory synapses, processes were classified as modulatory if they contained mostly large vesicles (>60 nm), either dark or clear. We distinguished two types of neuromodulatory processes (see Results; *Figure 9*). First, the AF with a distinctive morphology - a menorah–shaped tree climb from the neuropil to the cortex while intermingling with SAM trunks. The AF displayed boutons filled either with dense core vesicles with an average size of 74.9 ± 1.6 nm (AF-Nm) or a mix of smaller vesicle types that could be round, flattened, or irregularly shaped (AF-mix). Various subtypes of AF-mix were observed. The AF-Nm neurons were heavily innervated by the SAMs without any sign of presynaptic specialization. In contrast, the AF-mix was innervated to a lesser extent by the SAMs and displayed some presynaptic sites as well as occasionally reciprocal contacts.

The second type was NMs which were dispersed more globally in the VL neuropil and within the SFL tract and lacked a general orientation. Several subtypes were characterized by the homogeneity of the vesicle types found along their length. Most of their boutons lacked any sign of synaptic connection but the few pre- or postsynaptic sites observed were unorthodox. These NM processes displayed the largest vesicles in the entire dataset with diameters up to 200 nm (mean ± SD; 96.3 ± 3.5 nm) within their bouton and an accumulation of much smaller vesicles was observed at their rather rare presynaptic sites.

### Glia

Cells were classified as glia if they showed clearly distinguishable gliofibrils (*Gray and Young, 1964*) and/or did not receive any synapses. At least two types of protoplasmic glia were identified (*Figure 1— figure supplement 1C*). (1) radial glia (*Young, 1971*) were full of gliofibrils and characterized by long radial processes and arborizations infiltrating the neuronal cell cortex. These cells appear to isolate neuronal types from each other. (2) The astrocyte-like glia had many gliofibrils within their processes, which showed an increased tortuosity densely infiltrating the SFL tract. Here they appeared to wrap distinct synaptic areas. Through repeating folds, this cell type appears to maximize self-contacts characterized by the apposition of their membranes and accumulation of electron-dense materials on both sides, possible sites for protein exchange. One astrocyte-like cell with its cell body located near a cell island was partially reconstructed; it covered an extensive area within the SFL tract. (3) Fibrous glia with cell bodies in the inner cortex were also observed; these did not receive any synapses in the reconstructed volume.

### Others

Cells were classified as others only if their cell body was reconstructed (*Figure 1—figure Supplement 1B*). These cells were nested in each other in the inner cortex forming a type of horizontal layer. There were at least two types of cells in this arrangement. First, the base was formed by elongated cell bodies whose nuclei showed bumpy irregular membranes often with nuclear folds. These cell bodies appeared to have very small projections, and, like the glia, they lacked synaptic connections. Above these elongated cells were more classically spherical cell bodies which appeared somewhat 'immature.' Their trunks, extending radially into the neuropil, often lacked synaptic connection and terminated early with filopodia typical of axon growth cones. Their fibers were rich in grayish filaments that could be filamentous actin, suggesting that this may be a site of plastic reorganization. Because of the similarity of this arrangement with neurogenic niches observed in the mammalian dentate gyrus of the hippocampus and the subventricular zone, these cells resemble 'progenitor-like cells.' We saw no sign of cell division.

## Identification and analysis of synapses

Chemical synapses were identified by features common to invertebrate synapses - the presence of a cluster of synaptic vesicles or 'cloud' associated with an active zone near the presynaptic membrane (*Gray, 1970*; *White et al., 1986*; *Ryan et al., 2016*; *Zheng et al., 2018*; *Meinertzhagen, 2019*; *Witvliet et al., 2021*). Although the active zone in our dataset was not as electron-dense as in other invertebrate systems, a pale active zone was still identifiable through careful observation. Synapses were annotated if the accumulation of synaptic vesicles persisted through at least 3 sections in a row (90 nm). We distinguished between traditional chemical synapses (fast transmission) with 30–60 nm synaptic vesicles and synapses of modulatory neurons with mostly large vesicles (*Witvliet et al., 2021*). Consistent with previous studies (*Gray and Young, 1964*; *Gray, 1970*), we did not find evidence of electrical connections (gap junction) or any clear postsynaptic density (PSD) at any chemical synapse within the vertical lobe. However, in the connectomics dataset, we observed a rather global membrane thickening at synaptic sites. This may be partly due to a lack of resolution and the observed global overstaining.

The high magnification dataset was used to confirm the synaptic nature of the connections observed in the connectomics dataset and the absence of PSD. The absence of PSD made it difficult to annotate polyadic varicosities (one cluster of vesicles could be associated with two postsynaptic targets). In ambiguous cases only the most likely partners were identified (larger surface contact), leading to a likely underestimation of the overall number of synapses. Because most of the cells were not fully contained in the reconstructed volume, it was not possible to assess the full collection of synapses between a given presynaptic and postsynaptic neuron. For this reason, we refer to a 'synaptic site,' a single anatomical release site or a single synapse. Multiple synaptic sites from the same cell were classified as a single input. Additionally, because cells had different lengths of arbor in the reconstructed volume, we expressed most of the results as a synaptic fraction and number of synaptic sites per 100 μm.

Although multiple annotators would have been ideal, due to project constraints and the difficulty of training people in an entirely new dataset where most synapses were being described for the first time, all synaptic annotations were made by a single annotator, giving consistency across annotations.

## Clustering of cells based on connectivity

As described above, the reconstructed cells and processes of the vertical lobe were classified based primarily on morphological criteria. We combined this classification with unsupervised clustering to obtain a visual summary of the connectivity patterns found in the lobe. First, we produced a cell-type wiring diagram connecting the 7 main cell types (*Figure 2*). The nodes of this graph comprise cell types, and the weighted edges between pairs of cell types represent the number of synapses between the processes that belonged to these cell types. We used a spring model to minimize the distances between the pairs of nodes (representing pairs of cell types) that corresponded to a relatively larger number of synapses (*Fruchterman and Reingold, 1991*). This diagram revealed that synapses within a cell type are rare in the vertical lobe (mainly in ascending fibers; see Results). In addition, visually inspecting this diagram clarified that the connectivity was highly structured and far from an 'all-to-all' connectivity. Each cell type was connected to a small number of other cell types (see Results). This observation led us to consider the common inputs and common outputs of specific processes (described below).

We also discovered that besides these uncommon within-type synapses, there were no connectivity cycles in the cell-type wiring diagram. This observation led us to analyze the possible cycles in the wiring diagram while considering individual neurites (see the above Criteria for cell classification). We used an unsupervised approach to visualize the connectivity among individual processes (*Figure 2—figure supplement 1*). This was done by considering the adjacency matrix $\mathbf{A_{NxN}}$ whose (**i, j**) entries represent the number of synapses from the presynaptic neuron **i** to the postsynaptic neuron **j,** considering **N** neurons (**N**=362). As specific groups of cells tended to synapse on other specific subsets of cells, a stereotypy observed from the cell-type diagram, we considered the number of shared inputs and shared outputs between individual processes, mathematically represented by the common output and common input symmetric matrices, **A\*A'** (**presynaptic**) and **A'\*A** (**postsynaptic**), respectively. The presynaptic matrix represents how many common targets exist for pairs of neurons, and the postsynaptic matrix represents how many common inputs

exist for pairs of neurons. To visualize the tendencies of neurons to share inputs or outputs, we considered the matrix **A\*A'+A'\*A** which sums for every pair of neurons the number of shared inputs and outputs, using subspace embedding (**Koren, 2005**) and 100 dimensions, heuristically picking a dimension that yielded sufficient separation between individual nodes (nearby dimension produced visually similar outcomes). This embedding of the neurons in the plane resulted in a close positioning with other neurons with a similarity of inputs and outputs (summed). The position of the entire layout and its orientation is arbitrary and not used to derive biological observations. To combine the cell type- and neuron-based diagrams, each neuron was located based on its location in the type-based wiring diagram, shifted by its location in the neuron-based wiring diagram (**Figure 2**).

## Analysis of feed-forwardness

The cell type-based wiring diagram analysis revealed that connectivity cycles were rare and neural activity propagated in only one direction in the VL, analyzed among populations of cells belonging to the seven cell types (**Figure 2**). To determine if there were cyclic synaptic information flows in the reconstructed circuit at the level of individual neurites, we used graph algorithms that detect directed cycles in directed graphs. Topological sorting algorithms (**Barth et al., 2004**) that determine a linear order in cycle-free directed graphs were used to detect cycles in the graph. A topological sorting is a sequence of nodes each of which is possibly connected only to nodes proceeding in the list. Hence, we inspected all the edges in the topological sorting that went backward. We found a few edges (n=4) that violated the forwardness of the circuit and each of these cases was related to a reciprocal synapse between a pair of nodes associated with a single synaptic site between the cells. We then calculated the probability of observing this level of forwardness in random graphs that agreed with the wiring diagram, i.e., having the same set of neurons and synapses as in the data, but in which the direction of synaptic transmission was randomized (n=10,000 random connectomes). This showed that the level of forwardness in the data is unlikely to occur by chance ($p < 0.05$).

## Classification of cell types from cell body morphology

Our reconstruction revealed that the SAM, CAM, and LN cell bodies lay in the VL cortex along with several other cell types: glia, others, and mix cells (Others; **Figure 1—figure supplement 1**). We noticed, consistently with early studies, that the LNs had extremely large cell bodies and that the newly discovered CAMs tended to have slightly larger cell bodies than the SAMs. We tested whether these observations could be used to classify cell types directly from the morphology of their cell bodies. We used 110 cells whose somata were contained in the volume and their processes were distinctly characterized in the neuropil by an expert annotator: 7 LNs (6.36%), 69 SAMs (62.73%), 16 CAMs (14.55%) and 18 others (16.36%). The selection of this subset of cells was agnostic to the details of the classification process described here. For each of these cells, we considered the volume of the cell body as the number of annotated voxels multiplied by the volume unit of a voxel, and the volume, surface area, and sphericity (**Wadell, 1935**) of the ellipsoid best matching the geometry of the cell body. Ellipsoid approximations were computed from the eigenvectors and eigenvalues of the covariance matrix of the voxel coordinates of the cell bodies.

Our tests showed that the information gained from these ellipsoids was sufficiently discriminative and hence the three parameters volumes, surface area, and sphericity were used for further analyses. Support Vector Machines (SVMs) were used to train, cross-validate and predict the three main cell classes found in the cortex: LN, SAM and CAM, and a 4th class, Other, to represent the seemingly small number of other cell types in the cortex (including glia, progenitor cells and possibly other cells of scarce representation). The min, max, mean and SD of these parameters were computed for each class as well as the training accuracy (see Results).

In addition, a set of 47 cells (each cell was annotated twice) was used to determine the accuracy of the SVMs and for leave-one-out cross-validation (see Results). Once the SVM model was determined from the training set, it was applied to a test set, comprising 1447 cells, that were densely annotated in the cortex (using mEMbrain and VAST as described above). The number and percentage of each of the four classes were calculated (see Results; **Figure 4**).

### Wire-length calculation

To calculate the length of the processes (see Results; *Figure 2*), we used a thinning algorithm to map each volumetric object into a skeleton and used graph theory algorithms to measure the total length of skeletons. This was done by calculating the graph distance between skeleton nodes representing the end of the volumetric object (a leaf; a node that is connected to at most one other node, e.g. of the last node in a branch terminating at a synapse) and skeleton nodes representing branching (nodes that are connected to more than two other nodes, e.g., a branching point of a dendrite-like structure or axonal branching). The distances between adjacent skeletal nodes were defined using the Euclidean distance between the voxels in the 3D 26-neighb0rhood mask (a 3 × 3 cube around each voxel). We then summed up all these distances to measure the wire length of each process. Occasionally the thinning algorithm produces a biologically meaningless cycle due to topological holes in the annotated objects (either biological or imperfect annotation). We guaranteed that only the shortest paths between special skeletal nodes (bifurcations and leaves) were used for the wire-length calculation using minimum distance graph algorithms between leaf and bifurcation nodes.

### SFL bouton volume calculation

Synaptic boutons are morphologically easy to identify, and they were accordingly individually segmented to allow volumetric measurements (see Results; *Figure 3*). Volumes of boutons were estimated from the voxel counts of each bouton. A *post hoc* exact permutation test for independent samples compared the volumes of the SFL SB and SFL LB. The p-values were adjusted according to the sequential Bonferroni correction as described by *Holm, 1979*.

### Subcellular reconstruction

The geometry of the SFL large synaptic boutons and the postsynaptic structures they innervate were reconstructed at the subcellular level using the high magnification dataset (see Results; *Figure 5*). Neuronal elements of interest were manually segmented on VAST. The labeled images and metadata containing the labels were exported and processed externally or accessed through the VAST API. Vesicles showing a clear synaptic center in each section were counted by outlining intracellular space containing vesicles (see more information below). Mitochondria were identified by the presence of an inner membrane with visible cristae. Transport endoplasm, large electron-lucent tubular organelles, spanned the reconstructed volume and appeared to be a continuous structure present throughout the length of the AM trunk. They showed large swellings at the presynaptic site, particularly at the 'palm.' Discontinuous thin lucent tubular organelles that sometimes appeared as dark lines were annotated as 'cisternae' (Cis). It is not clear whether the Cis (up to 15 observed in one palm) were part of the larger endoplasmic reticulum network or were isolated entities.

### Counting, modeling, and measuring vesicles

An annotator painted all individual vesicles of several compartments using distinct colors per compartment. Distinct colors were used to annotate the active zones. The center of mass and areas of the vesicle cross-sections were used to approximate the centers and spherical representations of individual vesicles, from which spherical 3D mesh representations were computed. Occasionally, several vesicles of the same compartment were overlapped spatially. This required us to use distance transforms and watersheds to achieve the separation of vesicles. Vesicle counts were made only after they were separated in space. Due to the section thickness (30 nm), most of the vesicles were clearly appreciated in a single section, whereas only the largest vesicles (>60 nm diameter) may have had a strong signal in two adjacent sections. The annotator attempted to avoid overrepresentation of a vesicle, when possible, but since this task is not unambiguous our counts likely represent an upper bound. Extremely small vesicle-like structures were not annotated since it was assumed that these structures were likely represented in adjacent sections.

### Rendering

For the rendering of all the traced images, 3D surface meshes of labeled objects were generated from VAST using VastTools (written in MATLAB) and imported into 3D Studio Max 2020 (Autodesk Inc) to generate 3D renderings of all the traced objects.

## Acknowledgements

This work was supported by Human Frontier Science Program (HFSP) grant no. RGP0042/2019-102, Israel Science Foundation (ISF) grant no. 1928/15, National Institutes of Health (NIH) grant no. 5U24 NS109102 and NIH U01 NS108637. This work was also supported by a traveling fellowship from the Aharon and Ephraim Katzir Study Grants to F Bidel. We thank Prof. Tamar Flash for her contribution during the inception phase of the project. We thank Prof. Jenny Kien for editing the manuscript.

## Additional information

### Funding

| Funder | Grant reference number | Author |
|---|---|---|
| Human Frontier Science Program | RGP0042/2019-102 | Binyamin Hochner |
| Israel Science Foundation | 1928/15 | Binyamin Hochner |
| National Institutes of Health | 5U24 NS109102 | Jeff William Lichtman |
| National Institutes of Health | U01 NS108637 | Jeff William Lichtman |
| Aharon and Ephraim Katzir Study Grants | | Flavie Bidel |

The funders had no role in study design, data collection and interpretation, or the decision to submit the work for publication.

### Author contributions

Flavie Bidel, Conceptualization, Data curation, Formal analysis, Funding acquisition, Investigation, Visualization, Methodology, Writing – original draft, Project administration, Writing – review and editing; Yaron Meirovitch, Conceptualization, Data curation, Software, Formal analysis, Investigation, Methodology, Writing – original draft, Project administration, Writing – review and editing; Richard Lee Schalek, Elisa Catherine Pavarino, Software, Methodology; Xiaotang Lu, Investigation, Methodology; Fuming Yang, Adi Shaked, Investigation; Adi Peleg, Yuelong Wu, Software; Tal Shomrat, Supplementary electrophysiological experiment; Daniel Raimund Berger, Software, Visualization, Writing – review and editing; Jeff William Lichtman, Resources, Supervision, Funding acquisition; Binyamin Hochner, Resources, Supervision, Funding acquisition, Writing – original draft, Project administration

### Author ORCIDs

Flavie Bidel http://orcid.org/0000-0003-2129-5314
Yaron Meirovitch http://orcid.org/0000-0002-1946-8012
Fuming Yang http://orcid.org/0000-0001-7561-155X
Daniel Raimund Berger http://orcid.org/0000-0002-9677-6932
Jeff William Lichtman http://orcid.org/0000-0002-0208-3212
Binyamin Hochner http://orcid.org/0000-0002-9638-7320

### Ethics

All experimental animals were held and handle according to the guidelines for the EU Directive 2010/63/EU for cephalopod welfare in order to minimize the suffering and distress of the animals.

### Decision letter and Author response

Decision letter https://doi.org/10.7554/eLife.84257.sa1
Author response https://doi.org/10.7554/eLife.84257.sa2

## Additional files

### Supplementary files

• MDAR checklist

## Data availability

The 'connectome dataset' is publicly available using (browser-based) Neuroglancer at https://lichtman. rc.fas.harvard.edu/octopus_connectomes with 8 nm resolution as well as the connectomic segmentation of these data available in the same web-format.

The following dataset was generated:

| Author(s) | Year | Dataset title | Dataset URL | Database and Identifier |
|---|---|---|---|---|
| Bidel F, Meirovitch Y, Schalek RL, Lu X, Pavarino EC, Yang F, Peleg A, Wu Y, Shomrat T, Berger DR, Shaked A, Lichtman JW, Hochner B | 2022 | Octopus vulgaris vertical EMme | https://lichtman. rc.fas.harvard. edu/octopus_ connectomes | Octopus connectomes, octopus_connectomes |

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
