## [Editor Report]

In this important study, neural circuit architecture within a brain module for learning and memory in the octopus was mapped from volume electron microscopy. The acquisition of this pioneering data set was followed by a compelling analysis of the circuits supporting learning and memory, and therefore behavioural plasticity, on the basis of the newly mapped vertical lobe microcircuit and prior studies on long-term potentiation (LTP) in this animal. The data and findings establish an important point of comparison with analogous brain structures and memory mechanisms in other organisms, such as the vertebrate cerebellum and the arthropod mushroom body, and expand the set of known and mapped neural circuit architectures for learning and memory. The results will serve as the basis for further studies on the fascinating cephalopods and inspire further exploration in the design of artificial neural networks.

---

## [Decision Letter]

**Decision letter after peer review:**

Thank you for submitting your article "Connectomics of the Octopus vulgaris vertical lobe provides insight into conserved and novel principles of a memory acquisition network" for consideration by *eLife*. Your article has been reviewed by 3 peer reviewers, and the evaluation has been overseen by a Reviewing Editor and John Huguenard as the Senior Editor. The reviewers have opted to remain anonymous.

What a wonderful paper. We are very much looking forward to your revisions.

Essential revisions:

1) Please revise the claim that the connectome of a partial octopus brain vertical lobe gyrus is representative of all 5 gyri; there are published immunocytochemical evidence that suggests some gyri contain cell types not present in the others, and furthermore, the data sets presented here are limited to one of the gyri.

2) Please could you consider expanding the discussion, to further enlighten the reader with your thoughts on the purpose of the plasticity related to "self-LTP" and the SAM cell type, as well as contextualize the findings with what's known about the behaviors controlled by the vertical lobe. This would strengthen the manuscript and enable the reader to better place the significance and impact of the findings.

3) Careful proofreading would be most appreciated, as there are a number of typos to the point of being distracting.

4) Some figure legends would benefit from expanding the text that describes them. In general, it would be best if figure legends are as detailed as necessary for a reader to not have to rely on the main text. Space limitations aren't an issue.

Please see detailed reviewer comments below.

The reviewers thoroughly enjoyed reading your manuscript and have provided a number of detailed, constructive comments in the hope to assist you in strengthening the manuscript and clarifying its impact.

*Reviewer #1 (Recommendations for the authors):*

1) In general, the manuscript needs proofreading. I'm putting this in because it's an issue throughout to the point of being distracting. I hope the authors address this by carefully going through the whole manuscript, not just a few things that I was able to catch.

Figure 3: Some spelling/grammatical mistakes ("One SFL axons," "tiradic Motif").

Line 251: "relative aboundance the SAMs" should be "relative abundance of the SAMs".

Line 266-269: In one place it says the error on the validation set is smaller than 4.01%, another place 4.1%. In the Results, it says there are 47 cells used for validation, in the Methods it says 49.

Figure 4 caption: "cell body geometry roundish, relatively smaller…" Missing punctuation here?

Line 563: It's 50 billion granule cells, so it should be 50 x 10^9, not 10^12. And it's 50 billion neurons in the human cerebellum, not mouse.

Lines 612-613: There's a reference to Kandel et al. 2012, but no such reference in the bibliography. Same for Malenka 2003. I'm not going to go through every reference, but did the authors use a citation manager? If not, every reference must be checked by hand to make sure it's in the bibliography.

755: SMA should be SAM.

2) There are a lot of acronyms in this paper, and not all of them are necessary. There are already 7 cell type acronyms to keep in one's head, and I think these would be enough. A few examples that seem too much:

– LNp could just be "LN processes."

– MB is introduced as an acronym for the mushroom body even though it's only used 4 times.

– LB and SB are borderline but personally, I think they could just be called large and small boutons.

– In Figure 10, "OL" is used in the figure, presumably for the optic lobe, but the acronym is never defined.

– Sometimes "SFLs" is used but "SFL" is introduced as a brain area, so I would use "SFL axons" instead of "SFLs."

To help, full names along with acronyms, when possible, should be included in section titles, especially in the sections that focus on particular cell types.

3) When writing about LTP specificity and associativity in the Discussion, the terms should be defined here so the reader doesn't have to search through the references to understand the precise definition being used by the authors. Because the references are missing from the bibliography, I couldn't figure out what the definitions being used are. In general, I found this section rather technical. Related to a point from the public review, I would have hoped to see more discussion about exactly what signals are being used for supervision/"associativity," rather than solely molecular mechanisms.

4) The figures are pixelated. They'd look much better in a vector graphics format (with embedded bitmaps for images that are not vectors). If bitmaps must be used for the figures, export them with a higher resolution, but in general the first option always looks best.

Other comments:

Line 169: "the connectivity is strictly feedforward (p<0.05)." If a p-value is given here it should be described how it was attained. Later, it seems like the analysis is saying that the network is more feedforward than chance but not purely feedforward because of reciprocal synapses. If this is the case, then "strictly feedforward" is incorrect.

Lines 555-557: Unclear if "expansion ratio" here means the ratio of input neurons to connections, or input neurons to output neurons.

Figure 11: Why is "average SFL axonal length" written at the bottom of the figure? This number isn't referenced in the caption and doesn't seem relevant. Also, what is the 103 {plus minus} 73 μm distance meant to represent? In the caption, it says there are 1.03 {plus minus} 0.73 large boutons per 100 μm, which means that the typical distance between large boutons should be 100 μm divided by 1.03 {plus minus} 0.73, not multiplied by it. Also, the Σ in the figure isn't defined, I assume it just means the total number of large boutons per SFL axon.

Line 760-761: "However, the processing power provided by the association module is computationally more elaborated than a single neuron can provide." This claim needs to be supported/explained. It seems like it is contradicted by the sentences before and after it, which seem to be saying that the system is equivalent to a single neuron with a complex dendritic tree. If the SAM input/output function is linear, this is true, as seems to be one of the points of Shomrat et al. 2011.

*Reviewer #2 (Recommendations for the authors):*

Line 16: While this study focused on a small portion of the anterior part of the lateral gyrus of the vertical lobe only, this sentence needs to be rephrased to avoid misunderstanding.

Line 41: The vertical lobe of Octopus vulgaris is not exceptionally large, actually quite small at the volumetric aspect. Based on Maddock and Young 1987, the volumetric estimate of the VL in the young adult (mantle length 80 mm) was 6.67 mm^3, approximately 4 % of CNS volume. In addition, the relevant citation to refer to the neuron number (Young 1963) is needed here and in some sections below.

Line 78-80: Rephrase this sentence as this study only showed the results based on a small proportion of the lateral gyrus. Given incomplete data on the rest of the lateral gyrus as well as the other 4 gyri, it is unclear if the present findings can represent an entire VL.

Line 105-121: Figure 1A, Correct the body axis label of this sagittal section where the current anterior end needs to be corrected as posterior where the subpedunculate lobe is located.

Also, considering this sagittal slice located close to the lateral side of the CNS, the location of the inferior frontal lobe (IFL) in Figure 1A is thus arguable. Either the lateral superior frontal lobe or subvertical lobe would be a better fit than IFL.

At a quick glance at the link of the connectome dataset, numerous obvious "unusual or overshot" tracing can be easily spotted by human eyes, often appearing as a pair of narrow elongated colour-shaded patches nearby the dark boundary between two large structures such as the yellow/blue pair packed between the green and red large structure at the left upper panel of the first page while clicking the link. Notably, these elongated and narrow pairs of segmentation did not show any specific morphological features within these colour-shaded areas. Similar issues can be frequently found elsewhere. Are they some sort of artefacts due to semi-automatic tracing? How many of these biases could have been included in analyses? Any control to eliminate these false signals/artefacts? Does this occur in some specific groups of neurons?

Line 153: the subtitle needs to be rephrased to avoid a misleading impression of the VL connectome.

Line 205-210: It is unclear if the authors talked about the AF alone or an overall feedforward neural network.

Line 268-285: The authors revealed the ratio between SAMs (89.3%) and CAMs (1.6%) based on the 1472 cell bodies from the lateral gyrus of a mature Octopus vulgaris used in this study (lack of information on mantle length or body weight in the method). However, it is unclear if this octopus specimen was a similar size to the specimen used for estimates of neuron number in Young 1963. In addition, it is unclear if each gyrus contains "the same amount of neuron" and keeps similar wiring patterns among the 5 gyri. This incomplete dataset could lead to an overstatement of the findings. This entire paragraph needs either to be rewritten or moved to discussion to combine with Lines 545-553.

Line 561 and 563: Please make a consistent way to present the neuron number, correct this throughout the text.

Line 1542: where did these figures come from? Missing citations or methods?

*Reviewer #3 (Recommendations for the authors):*

I am very impressed and fascinated by this work.

I have a few recommendations concerning the readability of the manuscript.

1) As a general comment, I think that the figure design and labelling are not always intuitive. The labels in the main figures are very small and the authors work with unnecessary or non-intuitive abbreviations (e.g. 'pos' for postsynaptic in Figure 9b).

2) Please make sure that all illustrations are explained in the appropriate figure legends. As an example, the green shading in Figure 1c is not explained in the legend, however in comparable panels in Figure 2). To improve readability, I would recommend double-checking that everything shown in the figure is explained in the respective legend. I also think that the legends are a bit on the short side for the complexity of the data.

3) I very much enjoyed reading the discussion. I however think that the associative motifs could be made a bit clearer: especially in the figure, a more 'zoomed-in' schematic for where plasticity would occur (with NO and other factors incorporated) and how this would change information integration could be helpful.

4) I recommend illustrating the criteria for synapse identification more clear in the main figure. The authors do mention, e.g., clustered synaptic vesicles and show an example in Figure 1. However, the example in Figure 1 may not be ideal. While I can clearly see synaptic vesicles (that are also found in the SAM postsynapses), I have difficulties discriminating further features – how is the 'associated active zone' identified? What exactly is scored as a synapse or pre/post synapse?

5) Line 349 – Please specify: 'were occasionally observed'.

6) As the authors cannot identify postsynaptic densities, can they exclude SFL input from not traced neurons to SAMs?

---

## [Author Response]

Essential revisions:1) Please revise the claim that the connectome of a partial octopus brain vertical lobe gyrus is representative of all 5 gyri; there are published immunocytochemical evidence that suggests some gyri contain cell types not present in the others, and furthermore, the data sets presented here are limited to one of the gyri.

We added specifications in the summary and introduction to clarify that our results represent data from one of the Octopus VL lateral gyri. While earlier evidence indicated that the main features and synaptic motifs of the “fast transmission network” are similar across all five lobuli, the neuromodulatory networks that convey teaching signals likely differ. Therefore, we added a paragraph in the introduction (L106-113) and a Discussion section titled “How well does a partial connectome of a small portion of one VL lateral lobule represent the connectivity patterns across all five VL lobuli?” (L894-917) outlining what we believe is consistent across VL lobuli and what is likely lobule-dependent.

2) Please could you consider expanding the discussion, to further enlighten the reader with your thoughts on the purpose of the plasticity related to "self-LTP" and the SAM cell type, as well as contextualize the findings with what's known about the behaviors controlled by the vertical lobe. This would strengthen the manuscript and enable the reader to better place the significance and impact of the findings.

Following this comment, we have made some revisions to improve the clarity of the paragraph on self-LTP (L712-744). Additionally, we added two sections to provide context on how the VL circuitry enables the detection of visual stimuli and environmental cues (rewarding or punishing) to control the stereotypic attack behavior of the octopus. These sections are titled: "Perspective on the VL involvement in octopus associative learning" (L774) and "Perspective on the involvement of the ascending fibers (AF) and the neuromodulatory fibers (NM) in the supervision of learning" (L786).

3) Careful proofreading would be most appreciated, as there are a number of typos to the point of being distracting.

We carefully proofread the manuscript and bibliography before resubmission.

4) Some figure legends would benefit from expanding the text that describes them. In general, it would be best if figure legends are as detailed as necessary for a reader to not have to rely on the main text. Space limitations aren't an issue.

We revised all figure legends to provide as much information as possible.

Reviewer #1 (Recommendations for the authors):1) In general, the manuscript needs proofreading. I'm putting this in because it's an issue throughout to the point of being distracting. I hope the authors address this by carefully going through the whole manuscript, not just a few things that I was able to catch.Figure 3: Some spelling/grammatical mistakes ("One SFL axons," "tiradic Motif").Line 251: "relative aboundance the SAMs" should be "relative abundance of the SAMs".Line 266-269: In one place it says the error on the validation set is smaller than 4.01%, another place 4.1%. In the Results, it says there are 47 cells used for validation, in the Methods it says 49.Figure 4 caption: "cell body geometry roundish, relatively smaller…" Missing punctuation here?Line 563: It's 50 billion granule cells, so it should be 50 x 10^9, not 10^12. And it's 50 billion neurons in the human cerebellum, not mouse.Lines 612-613: There's a reference to Kandel et al. 2012, but no such reference in the bibliography. Same for Malenka 2003. I'm not going to go through every reference, but did the authors use a citation manager? If not, every reference must be checked by hand to make sure it's in the bibliography.755: SMA should be SAM.

We would like to express our gratitude to the reviewers for their valuable assistance in identifying typos and improving the bibliography. We have incorporated the feedback provided by reviewer #1 and carefully proofread the manuscript before resubmitting it.

2) There are a lot of acronyms in this paper, and not all of them are necessary. There are already 7 cell type acronyms to keep in one's head, and I think these would be enough. A few examples that seem too much:– LNp could just be "LN processes."

We replaced LNp with LN (or LN processes) throughout the manuscript.

– MB is introduced as an acronym for the mushroom body even though it's only used 4 times.

We do not use the acronym MB anymore.

– LB and SB are borderline but personally, I think they could just be called large and small boutons.

"LB" and "SB" are frequently used throughout the manuscript and allow for shorter annotations in figures. Additionally, the acronyms provide an identity to the large and small boutons. Therefore, we have decided to keep these two acronyms.

– In Figure 10, "OL" is used in the figure, presumably for the optic lobe, but the acronym is never defined.

We replaced “OL” with Optic lobes in figure 10.

– Sometimes "SFLs" is used but "SFL" is introduced as a brain area, so I would use "SFL axons" instead of "SFLs."

We replaced “SFLs” with “SFL axons” throughout the manuscript.

To help, full names along with acronyms, when possible, should be included in section titles, especially in the sections that focus on particular cell types.

Whenever possible, both the full name and acronym are now provided in the Results sections for each specific cell type as a reminder.

3) When writing about LTP specificity and associativity in the Discussion, the terms should be defined here so the reader doesn't have to search through the references to understand the precise definition being used by the authors. Because the references are missing from the bibliography, I couldn't figure out what the definitions being used are. In general, I found this section rather technical. Related to a point from the public review, I would have hoped to see more discussion about exactly what signals are being used for supervision/"associativity," rather than solely molecular mechanisms.

We thoroughly checked the bibliography and added appropriate references. We clarified and simplified the section on LTP specificity and associativity, opting for a balanced description, including the molecular aspects of these mechanisms.

4) The figures are pixelated. They'd look much better in a vector graphics format (with embedded bitmaps for images that are not vectors). If bitmaps must be used for the figures, export them with a higher resolution, but in general the first option always looks best.

We are thankful for this suggestion. Some of the figures embedded in the manuscript appeared quite pixelated. For this second submission, we saved all figures separately in vector graphics format (PDF). Plots and text should no longer appear pixelated. However, the 3D rendering resolution was previously optimized for practical work in Photoshop, and re-rendering these at this stage will be a large effort.

Other comments:Line 169: "the connectivity is strictly feedforward (p<0.05)." If a p-value is given here it should be described how it was attained. Later, it seems like the analysis is saying that the network is more feedforward than chance but not purely feedforward because of reciprocal synapses. If this is the case, then "strictly feedforward" is incorrect.

The reciprocal synapses are the exception and not the rule but for better clarity we changed “strictly feedforward” to “feedforward”. We added to the figure caption a short explanation of how the p-value was obtained, and more information is found in the methods section under the paragraph titled “Analysis of feed-forwardness”.

Lines 555-557: Unclear if "expansion ratio" here means the ratio of input neurons to connections, or input neurons to output neurons.

We explain better the meaning of “expansion ratio” in a new section discussing the relevance of the finding of a single SFL input to each SAM in the context of other memory-storing networks. See L607-635.

Figure 11: Why is "average SFL axonal length" written at the bottom of the figure? This number isn't referenced in the caption and doesn't seem relevant. Also, what is the 103 {plus minus} 73 μm distance meant to represent? In the caption, it says there are 1.03 {plus minus} 0.73 large boutons per 100 μm, which means that the typical distance between large boutons should be 100 μm divided by 1.03 {plus minus} 0.73, not multiplied by it. Also, the Σ in the figure isn't defined, I assume it just means the total number of large boutons per SFL axon.

For clarity, these distances have been removed from the figure. More details on this can be find L313.

Line 760-761: "However, the processing power provided by the association module is computationally more elaborated than a single neuron can provide." This claim needs to be supported/explained. It seems like it is contradicted by the sentences before and after it, which seem to be saying that the system is equivalent to a single neuron with a complex dendritic tree. If the SAM input/output function is linear, this is true, as seems to be one of the points of Shomrat et al. 2011.

As pointed out by the reviewer, this claim was confusing and was accordingly removed.

Reviewer #2 (Recommendations for the authors):Line 16: While this study focused on a small portion of the anterior part of the lateral gyrus of the vertical lobe only, this sentence needs to be rephrased to avoid misunderstanding.

We revised L17 to read as follows: "We present the first analysis of the connectome of a small volume of the Octopus vulgaris vertical lobe (VL).” While we do not include “lateral lobule” in the summary, it is now specified throughout the manuscript, with new clarifications in the introduction and a section in the discussion.

Line 41: The vertical lobe of Octopus vulgaris is not exceptionally large, actually quite small at the volumetric aspect. Based on Maddock and Young 1987, the volumetric estimate of the VL in the young adult (mantle length 80 mm) was 6.67 mm^3, approximately 4 % of CNS volume. In addition, the relevant citation to refer to the neuron number (Young 1963) is needed here and in some sections below.

We revised this sentence, now at L42, to read, "It contains 25x10^6^ neurons, about half the neurons in the central nervous system (Young 1963)." We have also added Young 1963 as a reference when discussing the number of cells in the VL, and again later in the discussion when mentioning the number of cells in the OL and SFL.

Line 78-80: Rephrase this sentence as this study only showed the results based on a small proportion of the lateral gyrus. Given incomplete data on the rest of the lateral gyrus as well as the other 4 gyri, it is unclear if the present findings can represent an entire VL.

The specification that the volume was from the lateral lobule was added to the sentence (L81). Additionally, we added a final sentence to discuss the possible translation of our results to the entire VL (L106-113). We also included a new Discussion section titled "How well does a partial connectome of a small portion of one VL lateral lobule represent the connectivity patterns across all five VL lobuli?” (L894-917) to address this aspect.

Line 105-121: Figure 1A, Correct the body axis label of this sagittal section where the current anterior end needs to be corrected as posterior where the subpedunculate lobe is located.

We corrected the body axis.

Also, considering this sagittal slice located close to the lateral side of the CNS, the location of the inferior frontal lobe (IFL) in Figure 1A is thus arguable. Either the lateral superior frontal lobe or subvertical lobe would be a better fit than IFL.

We understand the reviewer's concern regarding the location of the inferior frontal lobe

(IFL) in Figure 1A. However, we would like to emphasize that the structural organization of the SFL/IFL is highly characteristic, and we are confident that the lobe labelled "IFL" is not the subvertical lobe. While it is true that looking at one slice only makes it arguable whether the lobe labelled IFL could be the lateral superior frontal lobe, after examining a few initial slices of the start of the block, we are under the impression that the IFL was labeled correctly.

At a quick glance at the link of the connectome dataset, numerous obvious "unusual or overshot" tracing can be easily spotted by human eyes, often appearing as a pair of narrow elongated colour-shaded patches nearby the dark boundary between two large structures such as the yellow/blue pair packed between the green and red large structure at the left upper panel of the first page while clicking the link. Notably, these elongated and narrow pairs of segmentation did not show any specific morphological features within these colour-shaded areas. Similar issues can be frequently found elsewhere. Are they some sort of artefacts due to semi-automatic tracing? How many of these biases could have been included in analyses? Any control to eliminate these false signals/artefacts? Does this occur in some specific groups of neurons?

The yellow/blue pair mentioned here corresponds to the human annotator's labeling of the contact between the large presynaptic contact (yellow) of the presynaptic element (green -SFL large bouton) with the large post-synaptic contact (blue narrow segmentation) of the postsynaptic element (red – SAM palm). This labeling was not aided by any semiautomatic tracing.

Although we initially intended to label only synaptic contacts and used this segmentation to assess synaptic contact size, we abandoned this idea as it appeared biased, as mentioned by the reviewer. Specifically, in the case of the SFL-SAM synapse, although it can be easily and confidently annotated, it is nearly impossible to distinguish between the physical contact and synaptic contact of the two elements (due to overstaining and lack of PSD). Thus, the narrow pair of segmentation is nothing more than an indication of a synapse localization with no intention to precisely label the synaptic contact size. Therefore, the segmentation of the synapse was used only to count the number of pre or post-synaptic contacts for a specific cell but not to assess synaptic size.

We intend to add annotations to the color segmentation that can be viewed online to make it clear for the reader.

Line 153: the subtitle needs to be rephrased to avoid a misleading impression of the VL connectome.

As mentioned earlier, we agree that to uncover the subtleties of the VL networks, the connectome of other VL lobes will have to be produced. However, we believe that at this stage, the general network revealed in this study should be qualitatively similar between the VL lobuli, at least for the fast transmission networks, as all the synaptic motifs reported in our study were also reported elsewhere (Gray and our own work). We could not identify any synaptic motif that was not uncovered in our sample, suggesting that the difference is likely to be mainly quantitative or related to the neuromodulation network. Therefore, we believe that the subtitle "The VL connectome is characterized by two parallel and interconnected feedforward networks" can still stand, even if we only produce the connectome of a small portion of one lateral gyrus of the VL.

Line 205-210: It is unclear if the authors talked about the AF alone or an overall feedforward neural network.

This paragraph was clarified.

Line 268-285: The authors revealed the ratio between SAMs (89.3%) and CAMs (1.6%) based on the 1472 cell bodies from the lateral gyrus of a mature Octopus vulgaris used in this study (lack of information on mantle length or body weight in the method).

We added the body weight in the method section: body weight ≈130g (L1207). For this study, we intentionally used a small *O. vulgaris*.

However, it is unclear if this octopus specimen was a similar size to the specimen used for estimates of neuron number in Young 1963. In addition, it is unclear if each gyrus contains "the same amount of neuron" and keeps similar wiring patterns among the 5 gyri. This incomplete dataset could lead to an overstatement of the findings. This entire paragraph needs either to be rewritten or moved to discussion to combine with Lines 545-553.

While we appreciate the reviewer's concern about the potential limitations of our study, we believe that this paragraph belongs to the result section as it is. We aimed to provide an initial estimate of the number of neurons as a first step toward understanding the cell diversity within the VL – a required task given the limited knowledge on this topic and the newly discovered cell types (specifically CAMs). Even though we acknowledge the limitation of using a small volume of EM, especially when separated from other tools, it was the first method that uncovered the CAM cell type. Although the number of CAMs might differ across the VL lobuli and within a lobule, data suggest they are present in all lobes.

As Young did before, we also provide a rough estimate of the number of neurons per type in this yet relatively unexplored organism. We have used Young's estimate of 25 million cells as a reference scale to help the reader understand the ranges of numbers we are discussing. However, we understand the potential variability in neuron number and wiring patterns across the five lobuli of the VL.

Therefore, to address the concern raised by the reviewer, we added specific statements in the summary and introduction to emphasize that our estimates derive from a small portion of one Octopus VL lateral gyrus. We also added a section dealing with the similarities and differences expected across the lobuli of the VL in terms of connectivity. We hope this clarification will help the reader better interpret our findings in the context of the overall knowledge of the VL.

Line 561 and 563: Please make a consistent way to present the neuron number, correct this throughout the text.

Done, as requested.

Line 1542: where did these figures come from? Missing citations or methods?

The methodology for "Figure 9 —figure supplement 1" is associated with the "Nodebased agglomeration of 3D compartments" method paragraph. In L1357, it is stated that "This membrane-constrained ML-aided reconstruction was used to quickly reconstruct all the synaptic partners of one LN (Figure 7C) and one AF (Figure 9 —figure supplement 1)." This figure supplement is associated with Figure 9, as mentioned in the Figure 9 caption (L570) and in the discussion when referring to the columnar organization (L897).

Reviewer #3 (Recommendations for the authors):I am very impressed and fascinated by this work.I have a few recommendations concerning the readability of the manuscript.1) As a general comment, I think that the figure design and labelling are not always intuitive. The labels in the main figures are very small and the authors work with unnecessary or non-intuitive abbreviations (e.g. 'pos' for postsynaptic in Figure 9b).

Whenever possible, we increased the size of the labels. However, in many cases, it was not feasible. Additionally, we standardized the labeling across all figures and adjusted the labeling of Fig9b for "Post" instead of "pos."

2) Please make sure that all illustrations are explained in the appropriate figure legends. As an example, the green shading in Figure 1c is not explained in the legend, however in comparable panels in Figure 2). To improve readability, I would recommend double-checking that everything shown in the figure is explained in the respective legend. I also think that the legends are a bit on the short side for the complexity of the data.

We have proofread all figure legends, added necessary details and extended them where required.

3) I very much enjoyed reading the discussion. I however think that the associative motifs could be made a bit clearer: especially in the figure, a more 'zoomed-in' schematic for where plasticity would occur (with NO and other factors incorporated) and how this would change information integration could be helpful.

As suggested by the reviewer, we added a zoomed-in view to Figure 11 to highlight the plasticity site.

4) I recommend illustrating the criteria for synapse identification more clear in the main figure. The authors do mention, e.g., clustered synaptic vesicles and show an example in Figure 1. However, the example in Figure 1 may not be ideal. While I can clearly see synaptic vesicles (that are also found in the SAM postsynapses), I have difficulties discriminating further features – how is the 'associated active zone' identified? What exactly is scored as a synapse or pre/post synapse?

We added a detail on synaptic identification/annotation in the caption of Figure 1.

5) Line 349 – Please specify: 'were occasionally observed'.

This sentence is intended to acknowledge the existence of reciprocal synapses that were not directly identified by the seed cell analysis network. Since they were not observed in any of the 28 seed cells, no accurate count can be provided. To provide clarification for the reader, we modified the sentence in L395 as follows: "Although not identified by the seed cell analysis network, we occasionally observed reciprocal synaptic contacts between SFL and SAM, as well as AF and SAM within the dataset."

6) As the authors cannot identify postsynaptic densities, can they exclude SFL input from not traced neurons to SAMs?

Whenever we traced a SAM, we carefully annotated all its synapses along the way. Annotating synapses does not require tracing the pre- or postsynaptic cell elements of the SAM being reconstructed. Although PSDs were not observed, all synapses annotated in this study were clearly identified by the accumulation of synaptic vesicles along the presynaptic membrane across at least three sections (> 90 nm).

The identification of synaptic input from the SFL to the SAM palm is so characteristic (Figure 3B) that it was easy to identify. Hence, accordingly, we are confident that all SFL inputs of the fully reconstructed SAMs were accurately annotated (within the volume; some of the SAM’s neuritic ending may have been cut short). Synaptic inputs from the AF to the SAM are more discrete, and we cannot exclude a slight underestimation of these synapses.